# Training-Free Constrained Generation With Stable Diffusion Models

**Stefano Zampini**[*]
Polytechnic of Turin
stefano.zampini@polito.it

**Jacob K. Christopher**[*]
University of Virginia
csk4sr@virginia.edu

**Luca Oneto**
University of Genoa
luca.oneto@unige.it

**Davide Anguita**
University of Genoa
davide.anguita@unige.it

**Ferdinando Fioretto**[†]
University of Virginia
fioretto@virginia.edu

## Abstract

Stable diffusion models represent the state-of-the-art in data synthesis across diverse domains and hold transformative potential for applications in science and engineering, e.g., by facilitating the discovery of novel solutions and simulating systems that are computationally intractable to model explicitly. While there is increasing effort to incorporate physics-based constraints into generative models, existing techniques are either limited in their applicability to latent diffusion frameworks or lack the capability to strictly enforce domain-specific constraints. To address this limitation this paper proposes a novel integration of stable diffusion models with constrained optimization frameworks, enabling the generation of outputs satisfying stringent physical and functional requirements. The effectiveness of this approach is demonstrated through material design experiments requiring adherence to precise morphometric properties, challenging inverse design tasks involving the generation of materials inducing specific stress-strain responses, and copyright-constrained content generation tasks. All code has been released at https://github.com/RAISELab-atUVA/Constrained-Stable-Diffusion.

## 1 Introduction

Diffusion models have emerged as powerful generative tools, synthesizing structured content from random noise through sequential denoising processes [1, 2]. These models have driven significant advancements across diverse domains, including engineering [3, 4], automation [5, 6], chemistry [7, 8], and medical analysis [9, 10]. The advent of stable diffusion models has further extended these capabilities, enabling efficient handling of high-dimensional data and more complex distributions [11]. This scalability makes stable diffusion models particularly promising for applications in science and engineering, where data is highly complex and fidelity is paramount.

However, despite their success in generating coherent content, diffusion models face a critical limitation when applied to domains that require outputs to adhere to strict criteria. In scientific and engineering contexts, generated data must go beyond merely resembling real-world examples; it must rigorously comply with predefined specifications, such as physical laws, safety standards, or design constraints grounded in first principles. When these criteria are not met, the outputs may become unreliable, unsuitable for practical use, or even hazardous, undermining trust in the model's applicability. *Bridging this gap is crucial for realizing the potential of diffusion models in scientific applications.*

---

[*]Equal contribution.

[†]Contact author

39th Conference on Neural Information Processing Systems (NeurIPS 2025).

Recent research has reported varying success in *augmenting model training* with (often specialized classes of) constraints and providing adherence to desired properties in selected domains [12–14]. Many of these methods, however, are restricted to simple constraint sets or feasible regions that can be easily approximated, such as a simplex, L2-ball, or polytope. These assumptions break down in scientific and engineering tasks where constraints may be non-linear, non-convex, or even non-differentiable. Others are fundamentally limited, as training-time enforcement offers only distribution-level adherence to constraints rather than per-sample guarantees, even in convex settings, and cannot generalize to unseen or altered constraints without retraining. While *inference-time* enforcement methods address these shortcomings, these approaches modify the reverse processes in the original data space [15, 16]. This makes them incompatible with diffusion models like Stable Diffusion, which operate on learned lower-dimensional representations of the data. Latent-space variants have begun to appear, but they rely on special measurement operators [17–19] or on learned soft penalties [20, 21], thus limiting general applicability to the problems of interest in this work.

We address this challenge by integrating a proximal mappings into the reverse steps of pretrained stable diffusion models. The paper makes the following contributions: At each iteration, the generated latent is adjusted with a gradient descent step on the score field followed by a proximal update for constraint correction, all without retraining the network. When the constraint set is convex we prove that every iterate remains in the feasible region and that the Markov chain converges almost surely to a feasible point. The same algorithm, kept unchanged, extends to strongly non-convex constraints and even to constraints that can be checked only through a black-box simulator by estimating stochastic subgradients with finite differences. Experiments on **(i)** porous-material synthesis with exact porosity and tortuosity, **(ii)** meta-material inverse design that matches target stress-strain curves through a finite-element solver in the loop, and **(iii)** content generation subject to copyright filters show near-zero violations and state-of-the-art performance in constrained generation.

## 2 Preliminaries: Diffusion Models

**Score-Based Diffusion Models** [22, 23] learn a data distribution by coupling a noising (forward) Markov chain with a learned denoising (reverse) chain. Let $\{x_t\}_{t=0}^{T}$ be the sequence of diffusion states with $x_0$ drawn from the original data distribution $p_{\text{data}}(x_0)$. In the *forward process*, Gaussian noise is added according to a fixed variance schedule $\bar{\alpha}_t$, where $\bar{\alpha}_{t+1} \leq \bar{\alpha}_t$, so that the learned marginal $q(x^T)$ approaches $\mathcal{N}(0, I)$ as $t \to T$. A score network $s_\theta(x_t, t)$ is trained to learn the score function $s_\theta(x_t, t) = \nabla_{x_t} \log q(x_t|x_0) \approx \nabla_{x_t} \log p(x_t|x_0)$, where the approximation holds under the assumption that $q(x_t|x_0)$ is a close proxy for the true distribution $p(x_t|x_0)$. The network is trained by applying noise $\epsilon \sim \mathcal{N}(0, I)$ perturbations to the $x_0$, such that $x_t = \sqrt{\bar{\alpha}_t}x_0 + \sqrt{1 - \bar{\alpha}_t}\epsilon$, with the training objective minimizing the predicted error in the score estimate of the noisy samples:

$$\min_{\theta} \mathbb{E}_{t \sim [1,T], \ x_0 \sim p_{\text{data}}, \ \epsilon \sim \mathcal{N}(0,I)} \left[ \|s_\theta(x_t, t) - \nabla_{x_t} \log q(x_t \mid x_0)\|_2^2 \right]. \tag{1}$$

In the *reverse process*, the trained score network $s_\theta(x_t, t)$ is used to iteratively reconstruct data samples from the noise distribution $p(x_T)$. At each step $t$, the model approximates the reverse transition, effectively reversing the diffusion process to *sample* high-quality data samples. Notably, Denoising Diffusion Probabilistic Models have been shown to be mathematically equivalent [2, 23].

**Stable Diffusion.** In latent-diffusion variants such as *Stable Diffusion* [24, 25], the same scheme operates in a compressed latent space. The architecture uses an encoder-decoder pair $\mathcal{E}$ and $\mathcal{D}$, where the encoder $\mathcal{E}$ maps the high-dimensional image data to a latent space, denoted $z_t$, and the decoder $\mathcal{D}$ reconstructs the final image from the latent space after the diffusion model has operated on it. The training objective remains consistent with Equation (1), with the exception that noise is applied to $z_0 = \mathcal{E}(x_0)$ as opposed to directly to $x_0$ as in ambient space diffusion models. The latent diffusion model is thus trained to denoise over the latent space as opposed to the image space. Notice, however, that training the denoiser does not directly interact with the decoder, as the denoiser's loss is defined over the latent space and does not connect to the finalized samples. This consideration is relevant to the design choice taken by this paper in the proposed solution, discussed in Section 4. After iterative denoising, the final sample can be obtained by decoding $z_0$ with $\mathcal{D}$.

# 3 Projected Langevin Dynamics

Score-based diffusion models sample by running annealed Langevin dynamics, a form of Langevin Monte Carlo adapted to multiple noise levels. At each level $t$, one takes $M$ iterations of

$$\boldsymbol{x}_t^{(i+1)} \;=\; \boldsymbol{x}_t^{(i)} + \gamma_t \, \boldsymbol{s}_\theta\big(\boldsymbol{x}_t^{(i)}, t\big) + \sqrt{2\gamma_t}\boldsymbol{\epsilon}, \quad \boldsymbol{\epsilon} \sim \mathcal{N}(\boldsymbol{0}, \boldsymbol{I}),$$

where $\boldsymbol{s}_\theta(\boldsymbol{x}, t) \approx \nabla_x \log q(\boldsymbol{x})$ is the fixed, learned score and $\gamma_t$ is a step size. This update performs stochastic gradient ascent on the learned log-density $\log q(\boldsymbol{x})$ with added Gaussian noise [26]. This optimization is performed either directly [22] or as the corrector step within a predictor-corrector framework [23]; in both cases, the result is an approximate gradient ascent on the density function, subject to noise. Provided this understanding, Christopher et al. [15] showed that for a constraint set $C$, enforcing $\boldsymbol{x}_t^{(i+1)} \in C$ turns sampling into a series of constrained problems.

Specifically, this framing shifts the traditional diffusion-based sampling procedure into a series of independent, per-timestep optimization subproblems, each responsible for denoising a single step while enforcing constraints:

$$\boldsymbol{x}_t^{(i+1)} = \mathcal{P}_{\mathbf{C}}\left(\boldsymbol{x}_t^{(i)} + \gamma_t \nabla_{\boldsymbol{x}_t^{(i)}} \log q(\boldsymbol{x}_t|\boldsymbol{x}_0) + \sqrt{2\gamma_t}\boldsymbol{\epsilon}\right), \tag{2}$$

where the projection operator $\mathcal{P}_{\mathbf{C}}(\boldsymbol{x}) = \operatorname{argmin}_{\boldsymbol{y} \in \mathbf{C}} \|\boldsymbol{y} - \boldsymbol{x}\|_2^2$ returns the nearest feasible sample. Note that Langevin dynamic annealing occurs in an external loop decreasing the noise level across $t = T, \ldots, 1$, so that noise level $\boldsymbol{\epsilon}$ and step size $\gamma_t$ remain fixed inside each subproblem. As $t \to 0$ the noise vanishes and the Langevin dynamics converge to a deterministic gradient ascent on the learned density function. Note also that annealed Langevin dynamics is known to reach an $\varepsilon$-stationary point of the objective with high probability [27]. In diffusion processes, the negative learned density function, $-\log q(\boldsymbol{x}_t|\boldsymbol{x}_0)$, serves as the objective, so Equation (2) can be viewed as *projected gradient descent* on this landscape. The results of [27] extend to non-convex settings, giving further theoretical support to the constrained sampler adopted here.

While this approach is applicable when diffusion models operate across the image space, it cannot be directly adapted to the context of stable diffusion as $\mathbf{C}$ cannot be concretely represented in the latent space where the reverse process occurs. Other works have attempted to impose select criteria on latent representations, but these methods rely on learning-based approaches that struggle in out-of-distribution settings [20, 21], making them unsuitable for scenarios requiring strict constraint adherence. This limitation likely explains their inapplicability in the engineering and scientific applications explored by [13–15].

# 4 Latent Space Correction

Addressing the challenge of imposing constraints directly in the latent space hinges on a key insight: while constraints may not be representable in the latent domain, their satisfaction can be evaluated at any stage of the diffusion process. Indeed, the decoder, $\mathcal{D}$, facilitates provides a differentiable transformation from the latent representation to the image space, where constraint violations can be directly quantified. Therefore, when the constraint function is differentiable, or even if its violations can be measured via a differentiable mechanism, gradient-based methods can be used to iteratively adjust the latent representation throughout the diffusion process to ensure constraint adherence.

## 4.1 Imposing Ambient Space Constraints on the Latent

This section begins by describing how constraints defined in the ambient space can be imposed on the latent during the denoising process. First, following the constrained optimization framework discussed in Section 3, at each noise level $t$ we recast the reverse process for *Stable Diffusion* as the following constrained optimization problem:

$$\underset{\mathbf{z}_T, \ldots, \mathbf{z}_1}{\text{minimize}} \sum_{t=T, \ldots, 1} -\log q(\mathbf{z}_t|\mathbf{z}_0) \qquad \text{s.t.:} \quad \mathbf{g}(\mathcal{D}(\mathbf{z}_t)) = 0, \tag{3}$$

where $\mathcal{D}$ maps the latent representation $\mathbf{z}_t$ into its original dimensions and $\mathbf{g}$ is a differentiable vector-valued function $\mathbf{g} : \mathbb{R}^d \to [0, \infty]$ measuring the distance to the constraint set $\mathbf{C}$. At each iteration of the diffusion process, our goal is to restore feasibility with respect to $\mathbf{g}$.

As the constraint function can only be meaningfully represented in the image space, its gradients with respect to the latent variables are computed by evaluating the function on the decoded representation $\mathcal{D}(\mathbf{z}_t)$. This process is facilitated by the computational graph:

$$\mathbf{z}_t \leftarrow \mathcal{D}(\mathbf{z}_t) = \boldsymbol{x}_t \leftarrow \mathbf{g}(\boldsymbol{x}_t) = \inf_{\boldsymbol{y} \in \mathbf{C}} \|\boldsymbol{y} - \boldsymbol{x}_t\|. \tag{4}$$

This enables iterative updates that reduce constraint violations by backpropagating gradients from the constraint function directly to the latent representation as,

$$\nabla_{\mathbf{z}_t} \mathbf{g} = \left(\partial \mathcal{D} / \partial \mathbf{z}_t\right)^\top \nabla_{\boldsymbol{x}_t} \mathbf{g}.$$

Thus constraint information, computed where it is meaningful (image space), can be back-propagated through the frozen decoder to steer the latent variables. The method leaves the score network and the decoder unchanged, adds no learnable parameters, and enables feasibility enforcement at every step even when $C$ is non-convex or specified only by a black-box simulator.

## 4.2 Proximal Langevin Dynamics

The representation of latent diffusion as a constrained optimization task enables the application of established techniques from constrained optimization. To this end, this section discusses a generalization of the orthogonal projections used in projected Langevin dynamics, formulated through a *Proximal Langevin Dynamics* scheme. Let a constraint be encoded by a proper, lower-semicontinuous convex penalty $\mathbf{g} : \mathbb{R}^d \to [0, \infty]$ whose zero set coincides with the feasible region. After each noisy ascent step we apply a proximal map:

$$\boldsymbol{x}_t^{(i+1)} = \operatorname{prox}_{\lambda \mathbf{g}} \underbrace{\left( \boldsymbol{x}_t^{(i)} + \gamma_t \nabla_{\mathbf{z}_t^{(i)}} \log q(\boldsymbol{x}_t | \boldsymbol{x}_0) + \sqrt{2\gamma_t} \boldsymbol{\epsilon} \right)}_{\text{Langevin Dynamics Step}}, \tag{5}$$

with the proximal operator defined as:

$$\operatorname{prox}_{\lambda \mathbf{g}}(\boldsymbol{x}_t) = \arg\min_{\boldsymbol{y}} \left\{ \mathbf{g}(\boldsymbol{y}) + \tfrac{1}{2\lambda} \|\boldsymbol{y} - \boldsymbol{x}_t\|_2^2 \right\}. \tag{6}$$

This operator balances maintaining similarity to the updated sample and adhering to the constraint function $\mathbf{g}$ as weighted by hyperparameter $\lambda$. Choosing $\mathbf{g}$ as the indicator of a set, reproduces the familiar projection step introduced earlier, but the proposed Proximal Langevin Dynamics also accommodates non-smooth regularizers, composite penalties, and constraints specified only through inner optimization subroutines [28–30]. Thus, because proximal maps can be evaluated efficiently even for implicit or geometrically intricate constraints, Proximal Langevin Dynamics extends Langevin-based sampling to settings where explicit projections are impractical while preserving the convergence guarantees of the original scheme. The next results provides a characterization on convergence to **(i)** the constraint set and **(ii)** the original data distribution $p_{\text{data}}$.

**Theorem 4.1** (Convergence to the Constraint Set). *Let $C$ be non-empty and $\beta$-prox-regular in the sense of [31], Def. 13.27, the score network satisfy $\|\nabla_{\boldsymbol{x}_t} \log q(\boldsymbol{x}_t)\| \leq G$ (a standard consequence of the bounded-data domain after normalization), and $\mathcal{D}$ is $\ell$-Lipschitz such that $\|\nabla \mathcal{D}(\mathbf{z})\| \leq \ell$. Then, for positive step sizes $\gamma_t, \leq \frac{1}{2G^2}\beta$, the following inequality holds for the distance to $C$:*

$$\operatorname{dist}\left(\mathcal{D}(\mathbf{z}_t'), C\right)^2 \leq (1 - 2\beta'\gamma_{t+1}) \operatorname{dist}\left(\mathcal{D}(\mathbf{z}_{t+1}'), C\right)^2 + \gamma_{t+1}^2 G^2, \quad \text{(non-asymptotic feasibility)}$$

*where $\mathbf{z}_t'$ is the pre-proximal mapping iterate, $g$ is $L$-smooth, $\beta' = \beta/(\ell L)$, and $\operatorname{dist}(\mathbf{z}_t', C)$ is the distance from $\mathbf{z}_t'$ to the feasible set $C$.*

Theorem 4.1 illustrates that the distance to the feasible set $C$ decreases at a rate of $1 - 2\beta'\gamma_{t+1}$ at each step (up to an additive $\gamma_{t+1}^2 G^2$ noise). Thus, the iterates converge to an $\epsilon$-feasible set in $\mathcal{O}(\frac{1}{\gamma_{\min}} \log(\frac{1}{\varepsilon}))$ steps, with $\gamma_{\min} = \min_t \gamma_t$. This is experimentally reflected in Section 6.1, where $C$ is convex and *all samples* converge to the feasible set.

**Theorem 4.2** (Training Distribution Fidelity). *Suppose the assumptions stated in Theorem 4.2 are satisfied. Then, for positive step sizes $\gamma_t, \leq \frac{1}{2G^2}\beta$, the following inequality holds for the Kullback-Leibler (KL) divergence from the data distribution:*

$$\operatorname{KL}\left(q(\mathbf{z}_{t-1}) \,\|\, p_{\text{data}}\right) \leq \operatorname{KL}\left(q(\mathbf{z}_t) \| p_{\text{data}}\right) + \gamma_t G^2, \tag{fidelity}$$

The inequality in Theorem 4.2 shows that the divergence from the training data distribution increases by at most $\frac{\beta}{2}$ (since $\gamma \leq \frac{1}{2G^2}\beta$) at each step and, consequently the cumulative divergence from the training distribution is $\mathcal{O}(\sum_t \gamma_t)$ and thus vanishes as $t \to 0$. Consequently, after at most $T^\star = \lceil \frac{1}{2\beta\gamma_{\min}} \log\left(\frac{\text{dist}(x_T, \boldsymbol{C})^2}{\varepsilon}\right)\rceil$ steps the expected constraint violation drops below $\varepsilon$ while their KL divergence from the original data distribution grows at most linearly in $\sum_t \gamma_t$ (negligible because $\gamma_t \to 0$ along the chain). This implies that our approach inherits the *same* sample quality guarantees as unconstrained diffusion, up to a tunable drift bounded by $\gamma_{\max} G^2$, (with $\gamma_{\max} = \max_t \gamma_t$), and, since $\gamma_t \to 0$ along the chain, this drift is negligible in practice. This result provides theoretical rationale for our method's comparable FID scores to unconstrained baselines for the image generation tasks in Sections 6.1 and 6.3. Proofs for both theorems are provided in Appendix H.

### 4.3 Training-Free Correction Algorithm

With the theoretical framework of our approach established, we are now ready to formalize the proposed training-free algorithm to impose constraints on $\mathbf{z}_t$. The algorithm can be decomposed into an outer minimizer, which corrects $\mathbf{z}_t$ throughout the sampling process, and an inner minimizer, which solves the proximal mapping subproblem. The former describes a high-level view of the entire constrained sampling process, whereas the latter presents the single-step sub-optimization procedure.

**Outer minimizer.** First, to impose corrections throughout the latent denoising process Equation (6) is modified to accommodate the $\mathcal{D}$ mapping:

$$\text{prox}_{\lambda \mathbf{g}}(\mathbf{z_t}) = \arg\min_{\boldsymbol{y}} \left\{ \mathbf{g}(\mathcal{D}(\boldsymbol{y})) + \tfrac{1}{2\lambda}\|\mathcal{D}(\boldsymbol{y}) - \mathcal{D}(\mathbf{z_t})\|_2^2 \right\} \tag{7}$$

At each sampling iteration, we first compute the pre-projection update $\mathbf{z}_t' = \mathbf{z}_t + \gamma_t \nabla_{\mathbf{z}_t} \log q(\mathbf{z}_t|\mathbf{z}_0) + \sqrt{2\gamma_t}\epsilon$ which incorporates both gradient and stochastic noise terms. We then apply a proximal operator to obtain the corrected latent $\hat{\mathbf{z}}_t = \text{prox}_{\lambda\mathbf{g}}(\mathbf{z_t}')$. This follows the Proximal Langevin Dynamics scheme outlined in the previous section.

**Inner minimizer.** Each proximal mapping is composed of a series of gradient updates to solve the outer minimizer. In practice, gradient descent is applied on the proximal operator's objective:

$$\mathbf{z}_t^{i+1} = \mathbf{z}_t^i - \nabla_{\mathbf{z}_t^i}\left[\mathbf{g}(\mathcal{D}(\mathbf{z}_t^i)) + \tfrac{1}{2\lambda}\|\mathcal{D}(\mathbf{z}_t^i) - \mathcal{D}(\mathbf{z_t^0})\|_2^2\right].$$

When a convergence criterion is met (e.g., $\mathbf{g}(\mathcal{D})(\mathbf{z}_t^{(i)}) < \delta$), the algorithm proceeds to the next denoising step. Algorithm 1 provides a pseudo-code for this gradient-based approach to applying the proximal operator within the stable diffusion sampling process.

In the case that $\mathbf{g}$ is an indicator function,

$$\mathbf{g}(\boldsymbol{y}) = \begin{cases} 0, & \boldsymbol{y} \in \mathbf{C}, \\ \infty, & \boldsymbol{y} \notin \mathbf{C}, \end{cases}$$

the proximal mapping reduces to a projection onto the constraint set. If $\mathcal{P}_{\mathbf{C}}$ can be formulated in the ambient space, the minimization can be fully imposed on the projection objective, $\|\mathcal{P}_{\mathbf{C}}(\mathcal{D}(\mathbf{z}_t)) - \mathcal{D}(\mathbf{z}_t)\|_2^2$. Notably, the gradients of this objective capture both the constraint violation term, $\mathbf{g}(\mathcal{D}(\mathbf{z}_t))$, and the distance term, $\frac{1}{2\lambda}\|\mathcal{D}(\boldsymbol{y}) - \mathcal{D}(\mathbf{z_t})\|_2^2$, within the prescribed tolerance, resulting in the solution to Eq. (7) at the end of the minimization. When $\mathbf{C}$ is convex, the projection can be constructed in closed-form (Section 6.1), but otherwise a Lagrangian relaxation can be employed [32]. For our experiments with non-convex constraint sets (Sections 6.2 and 6.3), we leverage an Augmented Lagrangian relaxation as described in Appendix D [33].

---

**Algorithm 1:** Sampler with Constraint Correction

**Input:** $\delta$ (violation tolerance), lr (learning rate)

**Define** $\texttt{prox\_objective}(\boldsymbol{x}_t^i)$:

    violation $\leftarrow \mathbf{g}(\boldsymbol{x}_t^i)$;

    distance $\leftarrow \frac{1}{2\lambda}\|\boldsymbol{x}_t^i - \boldsymbol{x}_t^0\|_2^2$;

    **return** violation + distance;

**for** $t \leftarrow T$ **to** $0$ **do**

    // Sampling steps (omitted).

    $i \leftarrow 0$;

    **while** $\mathbf{g}(\mathcal{D}(\mathbf{z}_t^i)) \geq \delta$ **do**

        $g \leftarrow \nabla_{\mathbf{z}_t^i} \texttt{prox\_objective}(\mathcal{D}(\mathbf{z}_t^i))$;

        $\mathbf{z}_t^{i+1} \leftarrow \mathbf{z}_t^i - (g \times \text{lr})$;   $i \leftarrow i + 1$;

**Output:** $\mathcal{D}(\mathbf{z}_0)$

---

*Importantly, in these cases the corrective step can be considered a projection of the latent under appropriate smoothness assumptions.* We justify this claim with the following rationale: under the assumption of latent space smoothness [34], the nearest feasible point in the image space closely corresponds to the nearest feasible point in the latent space. Therefore, while the distance component of the proximal operator, $\frac{1}{2\lambda}\|\mathcal{D}(\mathbf{z}_t^i) - \mathcal{D}(\mathbf{z}_t^0)\|_2^2$, is evaluated in the image space, minimizing this distance in the image space also minimizes the corresponding distance in the latent space, validating the use of this corrective step as a latent projector.

Finally, when **g** cannot be represented by a differentiable function, such as when the constraints are evaluated by an external simulator (as in Section 6.2) or when the constraints are too general to represent in closed-form (as in Section 6.3), it is necessary to approximate this objective to Equation (7) using other approaches. We discuss this further in the next section and empirically validate such approaches in Section 6.

## 5    Complex Constraint Evaluation

While in the previous section we discuss how to endow mathematical properties within stable diffusion, many desirable properties cannot be directly expressed as explicit mathematical expressions. Particularly when dealing with physical simulators, heuristic-based analytics, and partial differential equations, it becomes often necessary to either **(i)** estimate these constraints with surrogate models or **(ii)** approximate the gradients of black-box models. To this end, we propose two proxy constraint correction methods that leverage differentiable optimization to enforce constraints.

**Differentiable surrogates.** Surrogate models introduce the ability to impose soft constraints that would otherwise be intractable. Specifically, we replace $\mathbf{g}(\boldsymbol{x}_t)$, the constraint evaluation function used in the optimization process, with a constraint violation function dependent on the surrogate model (e.g., a distance function between the target properties and the surrogate model's predictions for these properties in $\boldsymbol{x}_t$ as shown in Section 6.3). This allows the surrogate to directly evaluate and guide the samples to adhere to the desired constraints at each step. Apart from this substitution, the overall algorithm remains identical to Algorithm 1. Through iterative corrections, the model *converges* to a corrected sample $\hat{\mathbf{z}}_t$ that satisfies the target constraints to the extent permitted by the surrogate's predictive accuracy. Appendix C provides an in-depth view of the differences of this approach with respect to classifier guidance.

**Differentiating through black-box simulators.** In some cases (including the setting of our meta-material design experiments in Section 6.2), a strong surrogate model cannot be leveraged to derive accurate violation functions. In such settings, the only viable option is to use a simulator to evaluate the constraints. However, these simulators are often non-differentiable, making it impossible to directly compute gradients for the proximal operator. To incorporate the non-differentiable simulator into the inner minimization process, this paper exploits a sensitivity analysis method inspired by the *differentiable perturbed optimizer (DPO)* adopted in the context of differentiable optimization [35, 36]. DPO introduces controlled perturbations to the optimization variables and smooths the resulting objective, yielding differentiable surrogate gradients. Following this idea, random local perturbations are injected into the simulator inputs and define a smoothed function $\bar{\phi}_\nu(\boldsymbol{x}_t) = \mathbb{E}_\epsilon[\phi(\boldsymbol{x}_t + \nu\epsilon)]$, where $\phi$ is the external simulator, $\epsilon \sim \mathcal{N}(0, I)$ is a random perturbation, and $\nu$ is a temperature parameter controlling the smoothing scale. An unbiased Monte Carlo estimate of $\bar{\phi}_\epsilon$ is obtained as

$$\bar{\phi}_\epsilon(\boldsymbol{x}_t) = \frac{1}{M}\sum_{m=1}^{M}\phi\Big(\boldsymbol{x}_t + \nu\epsilon^{(m)}\Big),$$

where $M$ is the number of perturbed samples. The gradient of this smoothed objective can be written as

$$\nabla_{\boldsymbol{x}_t}\bar{\phi}_\nu(\boldsymbol{x}_t) = \frac{1}{\nu}\mathbb{E}[\phi(\boldsymbol{x}_t + \nu\epsilon)\,\epsilon],$$

which corresponds to a finite-difference estimator whose scaling term $1/\nu$ is absorbed into the proximal step size during optimization. Finally, this smoothed estimator enables a differentiable loss formulation:

$$\nabla_{\boldsymbol{x}_t}\mathcal{L}(\phi(\boldsymbol{x}_t)) = -\big(\bar{\phi}_\epsilon(\boldsymbol{x}_t) - target\big),$$

which is used to update the latent variable $\mathbf{z}_t$ (*via* $\boldsymbol{x}_t = \mathcal{D}(\mathbf{z}_t)$) through proximal Langevin dynamics.

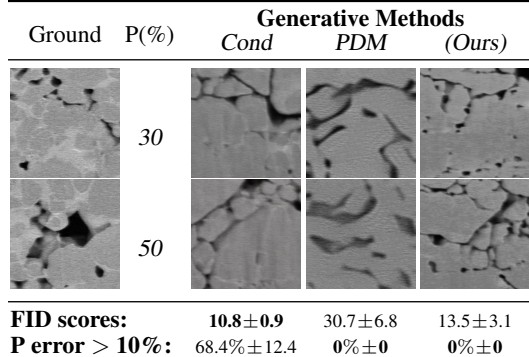

| Ground | P(%) | Generative Methods | | |
|---|---|---|---|---|
| | | *Cond* | *PDM* | *(Ours)* |
| | *30* | | | |
| | *50* | | | |
| **FID scores:** | | 10.8±0.9 | 30.7±6.8 | 13.5±3.1 |
| **P error > 10%:** | | 68.4%±12.4 | 0%±0 | 0%±0 |

Figure 1: Comparison of model performance in terms of FID score and constraint satisfaction (percentage of samples that does not satisfy the target porosity with a margin of 10%).

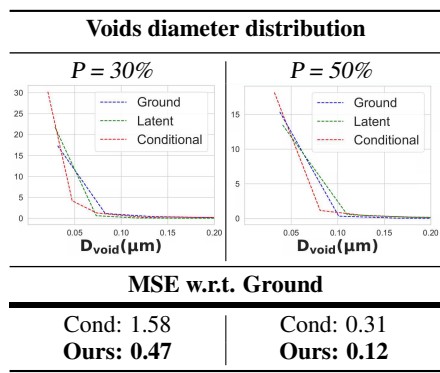

| **MSE w.r.t. Ground** | |
|---|---|
| Cond: 1.58 | Cond: 0.31 |
| **Ours: 0.47** | **Ours: 0.12** |

Figure 2: Distribution of void diameters in the training set (Ground) and in data generated by Conditional diffusion model and Latent Constrained Diffusion models.

## 6 Experiments

The performance of our method is evaluated in three domains, highlighting its applicability to diverse settings. Supplementary results and baselines are discussed in Appendices E and F.

**Baselines.** In each setting, performance is benchmarked against a series of baselines.

1. **Conditional Diffusion Model (Cond):** To assess the contribution of latent diffusion itself, we include a reference baseline consisting of an identical Stable Diffusion with text-conditioned constraints guidance [37].
2. **Projected Diffusion Models (PDM):** Following [15], this approach enforces feasibility by projecting onto the constraint set at each denoising step in the image space. PDM represents the current state-of-the-art for constrained generation in Sections 6.1 and 6.3.
3. **Bastek and Kochmann (2023):** For the task in Section 6.2, where PDM is inapplicable, we compare with this specialized method, which constitutes the state-of-the-art in that domain [38].

Collectively, these baselines capture the strongest existing constrained-generation methods, enabling a clear assessment of the improvements introduced by our latent constrained diffusion framework.

### 6.1 Microstructure Generation

Microstructure imaging data is critical in material science domains for discovering structure-property linkages. However, the availability of this data is limited on account of prohibitive costs to obtain high-resolution images of these microstructures. In this experiment, we task the model with generating samples subject to a constraint on the porosity levels of the output microstructures. Specifically, the goal is to generate new microstructures with specified, and often previously unobserved, porosity levels from a limited dataset of microstructure materials.

For this experiment we obtain the dataset used by [15, 39]. Notably, there are two significant obstacles to using this dataset: *data sparsity* and *absence of feasible samples*. To address the former limitation, we subsample the original microstructure images to generate the dataset using $64 \times 64$ images patches that have been upscaled to $1024 \times 1024$. To the latter point, while the dataset contains many samples that fall within lower porosity ranges, it is much more sparse at higher porosities. Hence, when constraining the porosity in these cases, often no feasible samples exist at a given porosity level.

**Inner minimizer.** To model the proximal operator for our proposed method, we use a projection operator in the image space and optimize with respect to this objective. Let $\boldsymbol{x}^{i,j}$ be the pixel value for row $i$ and column $j$, where $\boldsymbol{x}^{i,j} \in [-1, 1]$ for all values of $i$ and $j$. The porosity is then,

$$porosity = \sum_{i=1}^{n} \sum_{j=1}^{m} \mathbb{1}\big(\boldsymbol{x}^{i,j} < 0\big),$$

where $\mathbb{1}(\cdot)$ is the indicator function, which evaluates to 1 if the condition inside holds and 0 otherwise. We can then construct a projection using a top-k algorithm to return,

$$\mathcal{P}_{\mathbf{C}}(\boldsymbol{x}) = \underset{\boldsymbol{y}^{i,j}}{\arg\min} \sum_{i,j} \|\boldsymbol{y}^{i,j} - \boldsymbol{x}^{i,j}\| \qquad \text{s.t.} \qquad \forall\, \boldsymbol{y}^{i,j} \in [-1,1], \quad \sum_{i=1}^{n}\sum_{j=1}^{m} \mathbb{1}\left(\boldsymbol{y}^{i,j} < 0\right) = K$$

where $K$ is the number of pixels that should be "porous". Importantly, since the above program is convex, our model provides a certificate on the satisfaction of such constraints in the generated materials. We refer the interested reader to Appendix G for additional discussion.

**Results.** A sample of the results of our experiments is presented in Figure 1. Compared to the *Projected Diffusion Model (PDM)*, latent diffusion approaches show a significant improvement. Latent diffusion models enable higher-quality and higher-resolution images. The previous state-of-the-art PDM, which operates without latent diffusion, had an *FID more than twice as high as the models incorporating latent diffusion*. Alternatively, the *Conditional Diffusion Model*, utilizing text-to-image conditioning, reports the best average FID of 10.8 but performs poorly with regard to other evaluation metrics. Conditioning via text prompts proved unsuitable for enforcing the porosity constraints, and on average, *only 31.6% of the samples had a porosity error less than 10%,* indicating that this method lacks reliability in constraint satisfaction despite its ability to match the training distribution. Furthermore, as shown in Figure 2, the conditional model performs significantly worse than our method on producing realistic microstructures.

| Model | MSE [↓] | Fraction of physically invalid shapes [↓] |
|---|---|---|
| *Cond* | 7.1±4.5 | 55% |
| *Bastek and Kochmann* | 6.4±4.6 | 20% |
| *Latent (Ours)* | **1.4±0.6** | **5**% |

Figure 3: Compare MSE w.r.t. target stress-strain response and rejection rate of physically inconsistent shapes.

In contrast, our *Latent Constrained Model* exhibits the most optimal characteristics. The proposed method satisfies the porosity constraints *exactly*, achieves an excellent FID scores, and provides the *highest level of microstructure realism* as assessed by the heuristic-based analysis. This indicates that our approach effectively balances constraint satisfaction with high-quality image generation. *This is a significant advantage over existing baselines, as the method ensures both high-quality image generation and precise adherence to the physical constraints.*

### 6.2 Metamaterial Inverse Design

Now, we demonstrate the efficacy of our method for inverse-design of mechanical metamaterials with specific nonlinear stress-strain behaviors. Achieving desired mechanical responses necessitates precise control over factors such as buckling, contact interactions, and large-strain deformations, which are inherently nonlinear and sensitive to small parametric variations. Traditional design approaches often rely on trial-and-error methods, which are time-consuming and may not guarantee optimal solutions.

Specifically, our task is to generate mechanical metamaterials that closely match a target stress-strain response. From [38], we obtain a dataset of periodic stochastic cellular structures subjected to large-strain compression. This dataset includes full-field data capturing

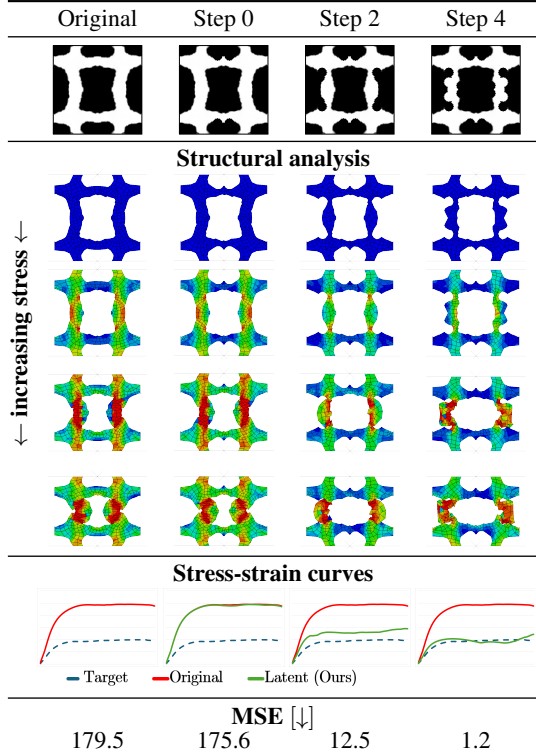

Figure 4: Successive steps of DPO. The sample is iteratively improved and the stress-strain curve aligns with the target. Structural analysis shows progressive deformation under controlled compression.

complex phenomena such as buckling and contact interactions. Because the problem is invariant with respect to length scale, the geometric variables can be treated as dimensionless. The stress is expressed in megapascals (MPa).

**Inner minimizer.** Exact constraint evaluation requires the use of *an external, non-differentiable simulator* $\phi$. Hence, we employ DPO as described in Section 5 to facilitate the proximal mapping with respect to the signals provided from this module. To compute the ground truth results for the stress-strain response, we employ Abaqus [40], using this simulator both for our correction steps and for validation of the accuracy of the generations. For this implementation, we set the number of perturbed samples $M = 10$, finding this provides strong enough approximations of the gradients to converge to feasible solutions.

**Results.** We illustrate the DPO process for our *Latent Constrained Model* in Figure 4. Firstly, note that our method facilitates the reduction of error tolerance in our projection to arbitrarily low levels. By reducing the tolerance and performing additional DPO iterations, we can tighten the bound on the projection operator, thereby enhancing its accuracy. Moreover, the integration of the simulator enables the model to generalize beyond the confines of the existing dataset (see Figure 7).

Due to the complexity of the stress-strain response constraints in this problem, other constraint-aware methods (i.e. Projected Diffusion Models) are inapplicable, and, hence, our analysis focuses on the performance of *Conditional Diffusion Model* baselines. We compare to **(1)** an unconstrained stable diffusion model identical to the one used for our method and **(2)** state-of-the-art method proposed by Bastek and Kochmann [38], which operates in the ambient space. While our approach optimizes samples to arbitrary levels of precision, we observe that these baselines exhibit high error bounds relative to the target stress-strain curves that are unable to be further optimized. As shown in Figure 3, with five DPO steps our method provides a *4.6x improvement* over the state-of-the-art model by [38] and a *5.1x improvement* over the conditional stable diffusion model MSE between the predicted structure stress-strain response and target response. *These results demonstrate the efficacy of our approach for inverse-design problems and generating samples that adhere to the target properties.*

### 6.3 Copyright-Safe Generation

Next, we explore the applicability of the proposed method for satisfying surrogate constraints. An important challenge for safe deployment of generative models is mitigating the risk of generating outputs which closely resemble copyrighted material. For this setting, a pretrained proxy model is fine-tuned to determine whether the generation infringes upon existing copyrighted material. This model has been calibrated so that the output logits can be directly used to evaluate the likelihood that the samples resemble existing protected material. Hence, by minimizing this surrogate constraint function, we directly minimize the likelihood that the output image includes copyrighted material.

To implement this, we define a permissible threshold for the likelihood function captured by the classifier. A balanced dataset of 8,000 images is constructed to fine-tune the classifier and diffusion models. Here, we use cartoon mouse characters 'Jerry,' from *Tom and Jerry*, and copyright-protected character 'Mickey Mouse'. When fine-tuning the diffusion model, we do not discriminate between these two characters, but the classifier is tuned to identify 'Mickey Mouse' as a copyrighted example.

**Inner minimizer.** Our correction step begins by performing Principal Component Analysis (PCA) on the 512 features input to the last layer and selecting the two principal components. This analysis yields two well-defined clusters corresponding to the class labels. Provided this, the inner minimizer differentiates with respect to a projection of the noisy sample onto the centroid of the target cluster, as illustrated in Figure 5 (top-right). Given the complexity of the constraints, we define a Lagrangian relaxation of the projection, terminating the iterative optimization based on proximity to the 'Jerry' cluster. Empirically, we found it was only necessary to trigger this proximal update if the classifier assigns a high probability to the sample being 'Mickey Mouse' at a given step.

**Results.** Figure 5 (bottom-right) reports the FID and constraint satisfaction, and Figure 5 (top-left) compares the evolution of the original sample and corrected sample. We implement a *Conditional Diffusion Model* baselines using and unconstrained stable diffusion model identical to the one used for our method. The conditional baseline generates the protected cartoon character (Mickey Mouse) *33% of the time*, despite conditioning it against these generations. The *Projected Diffusion Model* also struggles in this domain, particularly due to the image-space architecture's inability to handle high dimensional data as effectively as the latent models; it failed to generate reasonable samples at

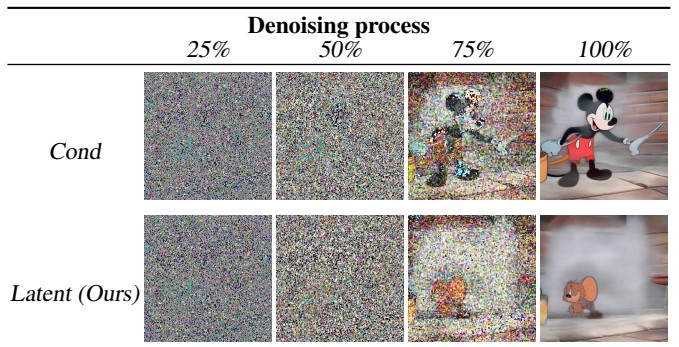
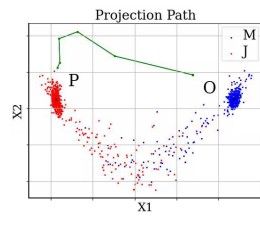

| | Denoising process | | | |
|---|---|---|---|---|
| | *25%* | *50%* | *75%* | *100%* |

*Cond*

*Latent (Ours)*

| Model | Constraint [↑] | FID [↓] |
|---|---|---|
| *Cond* | 67 % | **61.2** |
| *PDM* | 71 % | 75.3 |
| *Latent (Ours)* | **90 %** | 65.1 |

Figure 5: **Left:** Denoising process of Cond vs. Latent (Ours). Out method drives the denoising toward a copyright-safe image. **Top-right:** Showing projection from original (O) to projected (P) in the PCA-2 space. **Bottom-right:** Constraint satisfaction and FID scores.

a resolution of 1024×1024, unlike the latent models which maintained image quality at this scale. As a result, we were constrained to operate at a much lower resolution of 64×64 during sampling, subsequently relying on post-hoc upscaling techniques to reach the target resolution. This led to higher FID scores and only marginal improvement over the conditional model's constraint satisfaction. Conversely, our *Latent Constrained Model* only generates the protected cartoon character 10% of the time, aligning with the expected bounds of the classifier's predictive accuracy. Our method has proven to be highly effective because it preserves the generative capabilities of the model while imposing the defined constraints. The FID scores of the generated images remain largely unaltered by the gradient-based correction. *This demonstrates that our approach can selectively modify generated content to avoid copyrighted material without compromising overall image quality.*

## 7 Conclusion

This paper provides the first work integrating constrained optimization into the sampling process of stable diffusion models. This intersection enables the generation of outputs that both resemble the training data and adhere to task-specific constraints. By leveraging differentiable constraint evaluation functions within a constrained optimization framework, the proposed method ensures the feasibility of generated samples while maintaining high-quality synthesis. Experimental results in material science and safety-critical domains highlight the model's ability to meet strict property requirements and mitigate risks, such as copyright infringement. This approach paves the way for broader and more responsible applications of diffusion in domains where strict adherence to constraints is paramount.

## Contributions

FF and JC conceived the idea and developed the initial methods. SZ and JC implemented the algorithms, contributed to discussions, and refined the research direction. SZ conducted the experiments and analysis, while JC contributed to theoretical development and additional experiments. FF provided overall guidance and coordination. JC, FF, and SZ co-wrote the paper. LO and DA funded SZ's visit to FF's lab.

## Acknowledgments

The material is based upon work supported by National Science Foundations (NSF) awards 2533631, 2401285, 2334448, and 2334936, and Defense Advanced Research Projects Agency (DARPA) under Contract No. #HR0011252E005. The authors acknowledge Research Computing at the University of Virginia for providing computational resources that have contributed to the results reported within this paper. The views and conclusions of this work are those of the authors only.

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

# A Limitation

**Classifier-based constraints.** Section 6.3 motivates the use of classifier-based constraints for stable diffusion models. While we illustrate one potential use case of this approach, we hold that this can generalize to arbitrary properties that can be captured using a classifier. We defer more rigorous comparison to classifier guidance [2] and classifier-free guidance [41] for future work.

**Stable video generation.** While there many exciting applications for using this approach for scientific and safety-critical domains when generating data in the image space, many more applications will be enabled by extending this work to video diffusion models. While we compare to video diffusion baselines in Section 6.2, our training-free correction algorithm is only applied to stable image diffusion models. The introduction of temporal constraints over video frames holds significant potential that we plan to investigate in subsequent studies.

**Integration of external simulators.** This paper motivates future study of embedding non-differentiable simulators within generative process. Yuan et al. [42] previously proposed the inclusion of physics-based simulators to augment diffusion generations, but in their case, differentiability was not considered as their simulation used a reinforcement-learning environment to directly return a modified version of the noisy sample $x_t$. More often, external simulators are used to provide a measure of constraint satisfaction rather than to transform a sample. The techniques explored in this paper provide a vastly more general framework for incorporating these black-box simulators as a differentiable components of the sampling process and, hence, opens the door for the integration of increasingly complex constraints. Further use cases will be explored in consecutive works.

# B Related Work

**Conditional diffusion guidance.** Conditional diffusion models have emerged as a powerful tool to guide generative models toward specific tasks. Classifier-based [43] and classifier-free [41] conditioning methods have been employed to frame higher-level constraints for inverse design problems [3, 10, 38, 44] and physically grounding generations [5, 42, 45]. Rombach et al. extended conditional guidance to stable diffusion models via class-conditioning, allowing similar guidance schemes to be applied for latent generation. However, while conditioning based approaches can effectively capture class-level specifications, they are largely ineffective when lower-level properties need to be satisfied (as demonstrated in Section 6.1).

**Training-free diffusion guidance.** Similar to classifier-based conditioning, training-free guidance approaches leverage an external classifier to guide generations to satisfy specific constraints. Juxtaposed to classifier-based conditioning, and the method proposed in this paper, training-free guidance leverages *off-the-shelf* classifiers which have been trained exclusively on clean data. Several approaches have been proposed which incorporate slight variations of training-free guidance to improve constraint adherence [46–49]. Ye et al. compose a unified view of these methods, detailing search strategies to optimize the implementation of this paradigm. Huang et al. improve constraint adherence by introducing a "trust schedule" that increases the strength of the guidance as the reverse process progresses but remain unable to exactly satisfy the constraint set, even within the statistical bounds of the employed classifier. Importantly, training-free guidance approaches suffer from two significant shortcomings. First, this paradigm exhibits worse performance than classifier-based guidance as the off-the-shelf classifiers provide inaccurate gradients at higher noise levels. Second, like classifier-based guidance, these guidance schemes are ineffective in satisfying lower-level constraints

**Post-processing optimization.** When strict constraints are required, diffusion outputs are frequently used as initial guesses for a subsequent constrained optimization procedure. This approach has been shown to be particularly advantageous in non-convex scenarios where the initial starting point strongly influences convergence to a feasible solution [52]. Other methods incorporate optimization objectives directly into the diffusion training process, essentially framing the post-processing optimization steps as an extension of the generative model [53, 54]. However, these methods rely on a succinctly formulated objective and therefore often remain effective only for niche problems—such as constrained trajectory optimization—limiting their applicability to a wider set of generative tasks. Furthermore, post-processing steps are agnostic to the original data distribution, and, hence, the constraint correction steps often results in divergence from this distribution altogether. This has been empirically demonstrated in previous studies on constrained diffusion model generation [15].

**Hard constraints for generative models.** Frerix et al. [12] proposed an approach to impose hard constraints on autoencoder outputs by scaling the generated data so that feasibility is enforced, but this solution is limited to simple linear constraints. Liu et al. [16] introduced "mirror mappings" to handle constraints, though their method applies solely to familiar convex constraint sets. Given the complex constraints examined in this paper, neither of these strategies was suitable for our experiments. Alternatively, Fishman et al. [13, 14] extended the classes of constraints that can be handled, but their approach is demonstrated only for trivial predictive tasks with MLPs where constraints can be represented as convex polytopes. This confines their method to constraints approximated by simple geometric shapes, such as L2-balls, simplices, or polytopes. Coletta et al. [55] provide a generalization of constrained diffusion approaches for time-series problems, and, while these settings are not explored in our work, we take inspiration from their framing of diffusion sampling as a constrained optimization framework. Most similar to our work are Sharma et al. [56] and Christopher et al. [15], concurrent studies which propose projected Langevin dynamics sampling processes for constrained generation tasks. Sharma et al. [56] develop this approach for constrained graph generation, making this work not directly applicable to the tasks explored in this work. Christopher et al. generalize this sampling process for arbitrary constraint sets, but, like the other methods for hard constraint imposition discussed, their work is not extended to stable diffusion models.

**Stable diffusion for constrained settings.** Although inverse design has been extensively studied in diffusion models operating directly in image space, fewer studies have investigated the potential of stable diffusion models for inverse design, as these models have primarily been applied in commercial rather than scientific contexts. Song et al. [17] propose *ReSample*, which repurposed latent diffusion models as inverse-problem solvers by alternating a hard data-consistency projection and stochastic resampling steps returning the sample to the learned manifold. While applicable to image reconstruction with a fixed measurement operator, this approach cannot be adapted to handle open-ended constraint sets as explored in this work as it assumes knowledge of this operator for the data-consistency projection. More recently, Zirvi et al. [18] follow-up on this line of research, demonstrating improvement over [17] and [19], a method proposed concurrently to *ReSample*, but all three methods are limited in the requirement of a forward measurement operator which does not appear in our explored settings. This dependency narrows the scope of these inverse design methods, and this rigid definition of inverse design problems similarly restricts the applicability similarly presented methodology for diffusion operating in the image space.

Notably, a handful of other works explore constrained settings more generally without requiring dependency on the measurement operator. Engel et al. [20] proposes learning the constraints in the latent space via a neural operator, applying this methodology to auto-encoder architectures. Although effective for soft constraint conditioning, our attempts to adapt such approaches to latent diffusion were unsuccessful and such approaches can be viewed as a weaker alternative to classifier-based conditional guidance. Rather, conditioning approaches such as proposed by [24] have proven much stronger alternatives, leading to our selection of constraint-conditioned stable diffusion as a baseline throughout the explored settings. Finally, concurrent to our study, Shi and Bulling [21] proposes *CLAD* for imposing *learned* constraints on latent diffusion model generations, but, importantly, their study has very different objectives than our work. As opposed to our work, which enables imposition of hard constraints at inference time, *CLAD* learns latent vectors to statistically optimize feasibility for visual planning tasks. Importantly, this approach outputs symbolic action tokens as opposed to high-resolution pixel images, a distinctly disconnected modality from those explored in this paper.

## C    Comparison to Classifier Guidance

At first glance, this approach might appear similar to classifier-guided diffusion [43], as both rely on an external predictive model to direct the generation process. However, the two methods fundamentally differ in how the methods apply the model's gradient. Classifier-guided diffusion encourages similarity to feasible training samples, offering implicit guidance. In contrast, our approach provides *statistical guarantees as to constraint satisfaction within the confidence levels of the classifier,* providing a more direct and targeted mechanism for integrating constraints into the generative process.

**Classifier-based guidance.** Applies Bayesian principles to direct generation toward a target class $y$, based on the decomposition:

$$\nabla_{\boldsymbol{x}_t} \log p(\boldsymbol{x}_t \mid y) = \nabla_{\boldsymbol{x}_t} \log p(\boldsymbol{x}_t) + \nabla_{\boldsymbol{x}_t} \log p(y \mid \boldsymbol{x}_t) \tag{9}$$

This conditional generation incorporates a classifier $p(y \mid \boldsymbol{x}_t)$ into the sampling process. During generation, the model updates the noisy sample $\boldsymbol{x}_t$ by combining the standard denoising step with the classifier's gradient:

$$x_{t+1} = x_t + \epsilon \nabla_{x_t} \log p(x_t) + \sqrt{2\epsilon} + w \nabla_{x_t} \log p(y \mid x_t) \tag{10}$$

Here, the classifier's gradient $w \nabla_{x_t} \log p(y \mid \boldsymbol{x}_t)$ guides the denoising toward samples likely belonging to class $y$, with $w$ controlling the guidance strength.

**Training-free guidance.** Extends the principles of classifier-based guidance by leveraging pretrained, "off-the-shelf" classifiers to steer the generation process without requiring additional training. As with classifier-based guidance, the conditional generation incorporates a classifier $p(y \mid \boldsymbol{x}_t)$ into the sampling process. However, rather than training a custom classifier tailored to the diffusion model, this approach directly uses existing models to compute the guidance term. By decoupling the classifier from the diffusion model training, training-free guidance achieves flexibility and reusability, making it a practical choice for tasks where suitable pretrained classifiers are available.

**Surrogate constraint corrections.** Introduce a structured method to enforce class-specific constraints by adjusting samples at specific diffusion steps. In this approach, a surrogate model modifies the sample $\mathbf{z}_t$ to $\hat{\mathbf{z}}_t$ to meet the target constraints. These corrections can be introduced either at the beginning of the diffusion process, setting a strong initial alignment to the target class and then allowing the model to evolve naturally, or at designated points within the denoising sequence to enforce the constraints more explicitly at each selected step. In contrast, while classifier-based guidance and training-free guidance continuously integrate classifier gradients to steer generation toward the target class, surrogate constraint corrections offer discrete, targeted adjustments throughout the reverse diffusion process. This makes surrogate constraints particularly effective when strict adherence to certain class-specific conditions is necessary at particular stages of the generation process.

# D    Augmented Lagrangian Method

In experimental settings 6.2 and 6.3, a closed-form projection operator cannot be derived in the image space, and it becomes necessary to solve this subproblem through gradient-based relaxations. In these settings, we adopt the *Augmented Lagrangian Method* to facilitate this relaxation [57]. This method operated by restructuring a constrained optimization problem, in our case

$$\arg \min_{y \in \mathbf{C}} \|\boldsymbol{y} - \mathcal{D}(\mathbf{z}_t)\|$$

as a minimization objective which captures feasibility constraints through penalty terms. Expressly, the use of Lagrangian multipliers $\lambda = (\lambda_1, \ldots, \lambda_n)$ and quadratic penalty terms $\mu = (\mu_1, \ldots, \mu_n)$ form the *dual variables* used to solve the relaxation. Hence, the objective becomes:

$$\arg \min_{\boldsymbol{y}} \|\boldsymbol{y} - \mathcal{D}(\mathbf{z}_t)\| + \lambda \, \mathbf{g}(\boldsymbol{y}) + \tfrac{\mu}{2} \, \mathbf{g}(\boldsymbol{y})^2.$$

This problem provides a lower-bound approximation to the original projection. Its Lagrangian dual solves:

$$\arg \max_{\lambda, \mu} \Big( \arg \min_{\boldsymbol{y}} \|\boldsymbol{y} - \mathcal{D}(\mathbf{z}_t)\| + \lambda \, \mathbf{g}(\boldsymbol{y}) + \tfrac{\mu}{2} \, \mathbf{g}(\boldsymbol{y})^2 \Big).$$

Following the dual updates described by [33], we perform the following updates:

$$\boldsymbol{y} \leftarrow \boldsymbol{y} - \gamma \nabla_{\boldsymbol{y}} \mathcal{L}_{\mathrm{ALM}}(\boldsymbol{y}, \lambda, \mu), \tag{11a}$$

$$\lambda \leftarrow \lambda + \mu \, \tilde{\phi}(\boldsymbol{y}), \tag{11b}$$

$$\mu \leftarrow \min(\alpha\mu, \mu_{\max}), \tag{11c}$$

where $\gamma$ is the gradient step size, $\alpha > 1$ is a scalar which increases $\mu$ over iterations, and $\mu_{\max}$ is an upper bound on $\mu$. This drives $\boldsymbol{y}$ to satisfy $g(\boldsymbol{y}) \approx 0$ while staying close to $\mathcal{D}(\mathbf{z}_t)$.

---

**Algorithm 2:** Augmented Lagrangian

**Input:** $\boldsymbol{x}_t, \lambda, \mu, \gamma, \alpha, \delta$
$\boldsymbol{y} \leftarrow \mathcal{D}(\mathbf{z}_t)$
**while** $\tilde{\phi}(\boldsymbol{y}) < \delta$ **do**
    **for** $j \leftarrow 1$ **to** *max_inner_iter* **do**
        $\mathcal{L}_{\mathrm{ALM}} \leftarrow$
        $\|\boldsymbol{y} - \mathcal{D}(\mathbf{z}_t)\| + \lambda \, \mathbf{g}(\boldsymbol{y}) + \tfrac{\mu}{2} \, \mathbf{g}(\boldsymbol{y})^2$
        $\boldsymbol{y} \leftarrow \boldsymbol{y} - \gamma \nabla_{\boldsymbol{y}} \mathcal{L}_{\mathrm{ALM}}$
    $\lambda \leftarrow \lambda + \mu \, \tilde{\phi}(\boldsymbol{y})$;
    $\mu \leftarrow \min(\alpha\mu, \mu_{\max})$
**return** $\boldsymbol{y}$

---

# E    Extended Results

In this section, we include additional results and figures from our experimental evaluation.

## E.1    Microstructure Generation

**Additional baselines.** To supplement the evaluation presented in paper, we also implemented the following baselines:

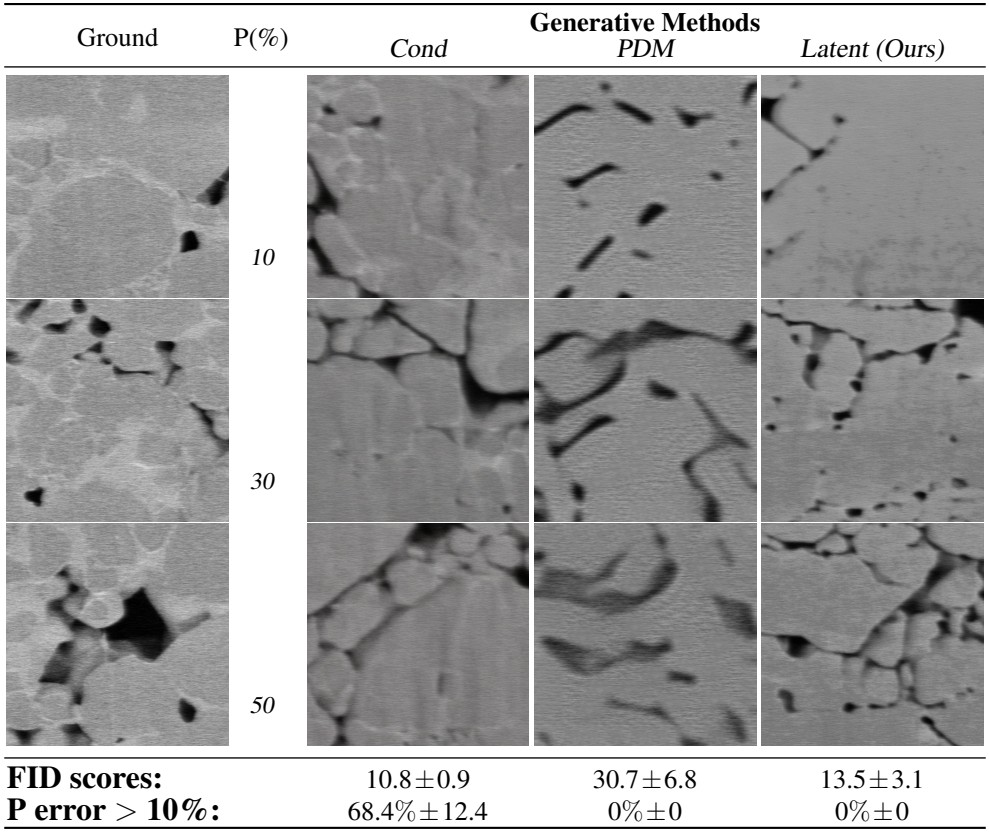

| Ground | P(%) | Generative Methods | | |
| --- | --- | --- | --- | --- |
| | | *Cond* | *PDM* | *Latent (Ours)* |
| | *10* | | | |
| | *30* | | | |
| | *50* | | | |
| **FID scores:** | | 10.8±0.9 | 30.7±6.8 | 13.5±3.1 |
| **P error > 10%:** | | 68.4%±12.4 | 0%±0 | 0%±0 |

Figure 6: Extended version of Figure 1

1. **Image Space Correction:** We implement a naive approach which converts the latent representation to the image space, projects the image, and then passes the feasible image through the encoder layer to return to the latent space.

2. **Learned Latent Corrector:** Adapting the implementation by [20] for diffusion models, we train a network to restore feasibility prior to the decoding step.

The *Image Space Correction* method, which involves re-encoding the image into the latent space after correcting it during various denoising steps, and the *Learned Latent Corrector* method, where a network is trained to project a latent vector toward a new state ensuring constraint satisfaction, both failed to produce viable samples. **Both baselines deviated significantly from the training set distribution,** resulting in high FID scores and generated images that lacked quality, failing to capture essential features of the dataset. Due to the inability of these methods to produce viable samples, we do not include them in Figure 1.

## E.2   Metamaterial Inverse Design

Figure 7 illustrates the performance of different models in interpolation (i.e., when the target curve falls within the stress range covered by the training set) and in extrapolation (i.e., when the target is outside this range). In addition to the proposed model, a Conditional Stable Diffusion model and a Conditional Video Diffusion model [38] are shown. The proposed model allows for arbitrarily small tolerance settings and outperforms the baselines in both tests.

Practically, one can select an error tolerance and compute budget for tailored for the specific application. Each iteration of the DPO necessitates approximately 30 seconds of computational time. Given our prescribed error tolerance, convergence is achieved within five iterations, culminating in a total computational duration of approximately 2.5 minutes per optimization run. Additionally, note that $\phi$ has not been optimized for runtime, operating exclusively on CPU cores.

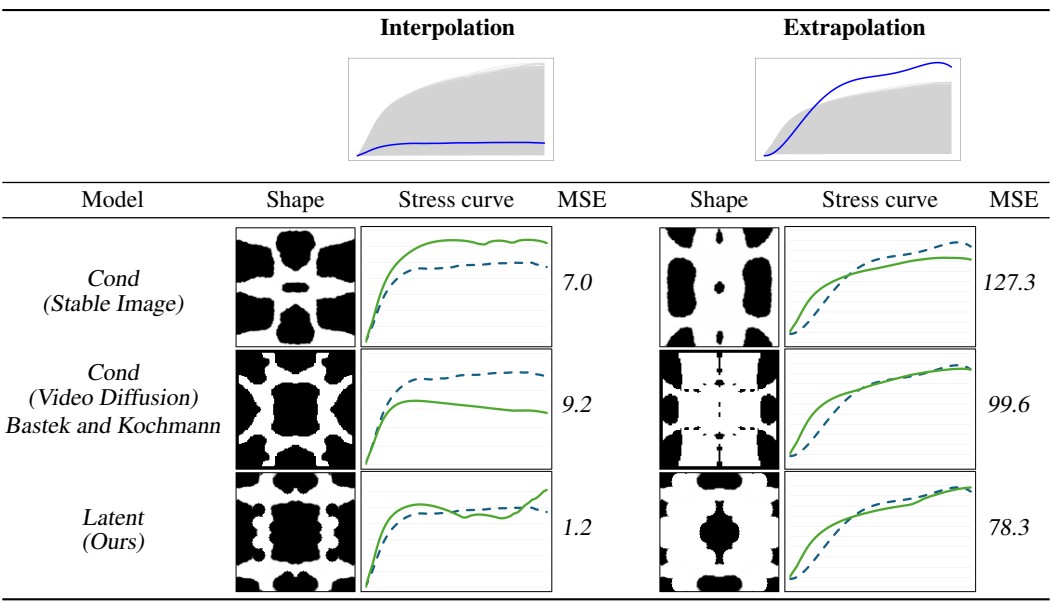

Figure 7: Performance of different models in interpolation and in extrapolation.

### E.3 Copyright-Safe Generation

As opposed to other approaches, the projection schedule is tuned differently for this setting. Particularly, it is only necessary to project during the initial stages of the generation. After this correction, the denoising process is allowed to evolve naturally without further intervention. This method ensures that the generated images are guided away from resembling copyrighted material while still allowing the model to produce high-quality outputs. By selectively modifying the generated content during the initial stages of denoising, we can effectively prevent the model from producing images that infringe on copyrights without significantly affecting the overall image quality.

**Surrogate implementation.** We begin by fine-tuning a classifier capable of predicting membership to one of two classes: 'Mickey Mouse' or 'Jerry'. The architecture of the classifier consists of a ResNet50 backbone, which is followed by two fully connected layers. These layers serve to progressively reduce the dimensionality of the feature map, first from 2048 to 512 and then from 512 to a single scalar feature, which represents the output of the classifier. A Sigmoid activation function is then applied to this final feature to estimate the probability that the input sample belongs to either the 'Mickey Mouse' or 'Jerry' class. This process ensures that the model outputs a value between 0 and 1, indicating the likelihood of each class membership. The classifier was evaluated on a held-out test set and demonstrated a strong performance, achieving an accuracy greater than 87%, which showcases its effectiveness in distinguishing between the two classes.

## F  Runtime

Note that the simulator employed in Section 6.2 has been specifically optimized for CPU execution; consequently, runtimes observed for these experiments are inherently longer compared to GPU-optimized procedures (refer to Table F). Importantly, optimizing computational runtime was not the primary focus of this study. Rather, the emphasis was placed on scientific discovery, prioritizing the fidelity and accuracy of the generated content. In the scientific domains explored, achieving exact adherence to physical constraints and established principles is crucial, as deviations would render the models unreliable and impractical for use.

A significant bottleneck frequently encountered in these experimental domains is the iterative cycle from experimental design to validation, which typically necessitates multiple trial-and-error iterations. Consequently, the capability to generate realistic synthetic materials exhibiting precisely controlled morphological or structural characteristics, as demonstrated at the fidelity levels reported herein, holds substantial potential for accelerating scientific workflows—from initial experimentation to full-scale production.

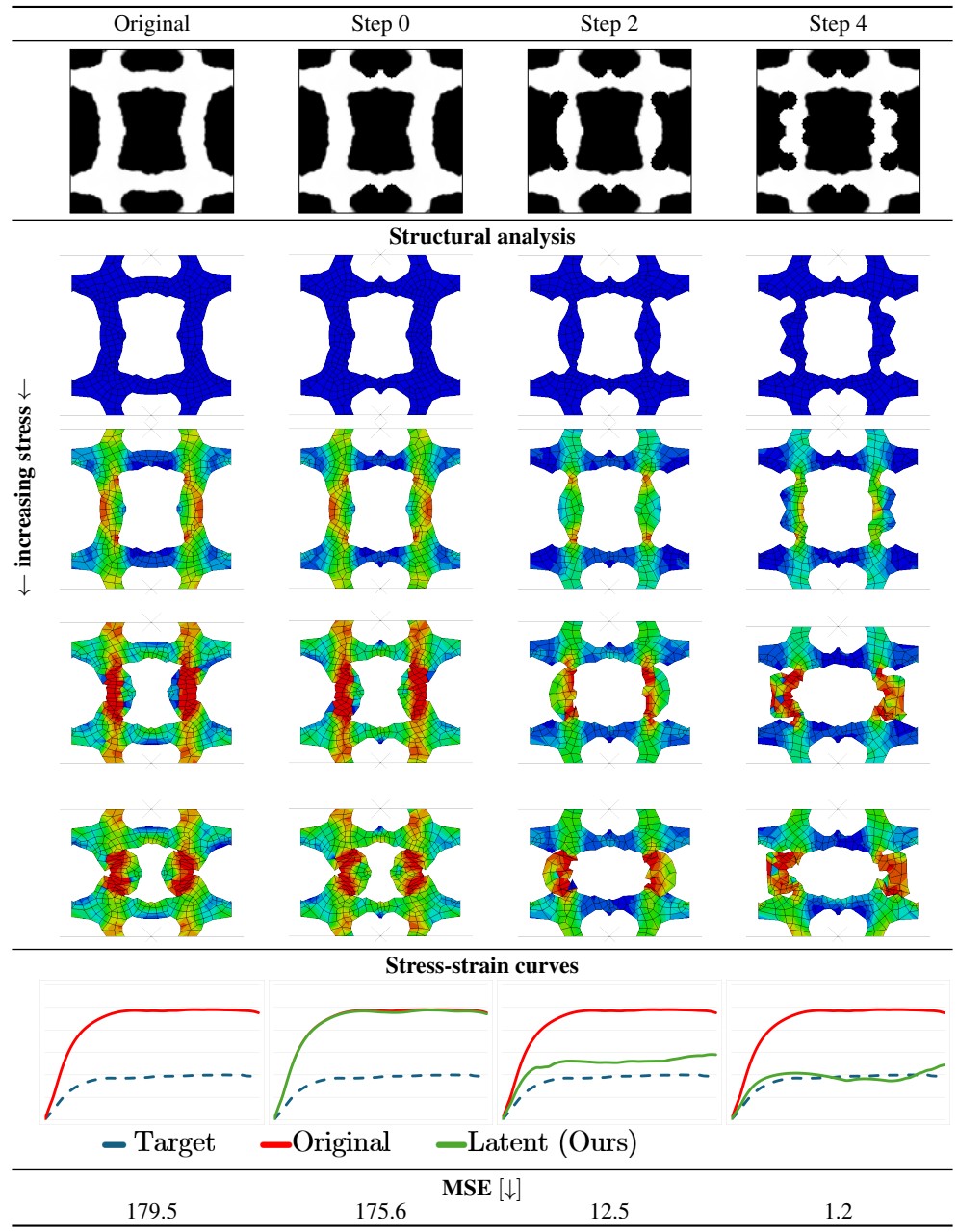

Figure 8: Enlarged version of Figure 4. Successive steps of DPO are shown. At each stage, $M$ perturbed shapes are generated, each undergoing structural analysis with $\phi$, which provides the corresponding stress-strain. The perturbation that produces the curve closest to the target is then selected, and a new perturbation-structural analysis-selection cycle begins, continuing until convergence is achieved. The convergence tolerance can be tightened as desired, provided it is compatible with the available computational cost.

| Experiment | Sampling Type | Time | Hardware |
|---|---|---|---|
| Microstructure | Conditional Sampling | 10 s | Nvidia A100-SXM4-80GB |
| Microstructure | Constrained Sampling | 50 s | Nvidia A100-SXM4-80GB |
| Metamaterials | Conditional Sampling | 5 s | Nvidia A100-SXM4-80GB |
| Metamaterials | Single Simulation | 30 s | Intel Core i7-8550U CPU |
| Copyright | Conditional Sampling | 10 s | Nvidia A100-SXM4-80GB |
| Copyright | Constrained Sampling | 65 s | Nvidia A100-SXM4-80GB |

Table 1: Experiment runtimes by sampling type and hardware.

## G  Feasibility Guarantees

First, note that when a projection (or approximation thereof) can be constructed in the image space, strict guarantees can be provided on the feasibility of the *final outputs* of the stable diffusion model. A final projection can be applied after decoding $\mathbf{z}_0$, and, as this operator is applied directly in the image space, constraint satisfaction is ensured if the projection is onto a convex set. As detailed in [57](§8.1), a projection of any vector $\boldsymbol{x} \in \mathbb{R}^n$ onto a non-empty closed convex set $\mathbf{C}$ exists and is unique.

**Theorem G.1.**  *Convex Constraint Guarantees: The proposed method provides feasibility guarantees for convex constraint.*

As we can derive a closed-form projection for convex sets in the ambient space after decoding our sample (e.g., in Section 6.1 where our proximal operator used is a projection), the proposed method thus provides guarantees for convex constraints.

## H  Supplementary Proofs

### H.1  Proof of Theorem 4.1 (Part 1): Non-Asymptotic Feasibility in the Image Space

*Proof.*  We begin by proving this bound holds for projected diffusion methods operating in the image space:

$$\text{dist}\left(\boldsymbol{x}'_t, \boldsymbol{C}\right)^2 \ \leq \ (1 - 2\beta\gamma_{t+1}) \ \text{dist}\left(\boldsymbol{x}'_{t+1}, \boldsymbol{C}\right)^2 + \gamma_{t+1}^2 G^2, \tag{12}$$

For clarity and simplicity of this proof, we omit superscripts in subsequent iterations, using only subscripts to denote both inner and outer iterations of the Langevin sampler. For instance, the sequence $\boldsymbol{x}_t^0, \ldots, \boldsymbol{x}_t^M, \boldsymbol{x}_{t-1}^0$ is now represented as $\boldsymbol{x}_t, \ldots, \boldsymbol{x}_{t-M}, \boldsymbol{x}_{t-(M+1)}$. Consequently, as annealing occurs at the outer loop level (every $M$ iterations), the step size $\gamma_t$ is no longer strictly decreasing with each iteration $t$, and instead satisfies $\gamma_t \geq \gamma_{t-1}$.

Consider that at each iteration of the denoising process, projected diffusion methods can be split into two steps:

1. **Gradient Step:** $\boldsymbol{x}'_t = \boldsymbol{x}_t + \gamma_t \underbrace{\nabla_{\boldsymbol{x}_t} \log q(\boldsymbol{x}_t)}_{s_t}$

2. **Projection Step:** $\boldsymbol{x}_{t-1} = \mathcal{P}_{\mathbf{C}}(\boldsymbol{x}'_t)$

These steps are sequentially applied in the reverse process to sample from a constrained subdistribution.

$$\boldsymbol{x}_t \to \overbrace{\boldsymbol{x}_t + \gamma_t s_t}^{\boldsymbol{x}'_t} \to \mathcal{P}_{\mathbf{C}}(\boldsymbol{x}'_t) = \boldsymbol{x}_{t-1} \to \overbrace{\boldsymbol{x}_{t-1} + \gamma_{t-1} s_{t-1}}^{\boldsymbol{x}'_{t-1}} \to \mathcal{P}_{\mathbf{C}}(\boldsymbol{x}'_{t-1}) = \boldsymbol{x}_{t-2} \ldots$$

By construction, $\boldsymbol{x}_{t-1} = \mathcal{P}_{\mathbf{C}}(\boldsymbol{x}'_t) \in \mathbf{C}$. Next, let us define the projection distance to $\mathbf{C}$ as:

$$f(\boldsymbol{x}) = \text{dist}(\boldsymbol{x}, \mathbf{C})^2 = \|\boldsymbol{x} - \mathcal{P}_{\mathbf{C}}(\boldsymbol{x})\|^2$$

Since $\mathbf{C}$ is $\beta$-prox regular, by definition the following hold:

- $f$ is differentiable outside $\mathbf{C}$ (in a neighborhood)

- $\nabla f(\boldsymbol{x}) = 2(\boldsymbol{x} - \mathcal{P}_{\mathbf{C}}(\boldsymbol{x}))$

- $\nabla f$ is $L$-Lipshitz with $L = \frac{2}{\beta}$

The "descent lemma" for $L$-smooth functions applies:

**Lemma H.1.** $\forall \boldsymbol{x}, \boldsymbol{y}$ *in the neighborhood of* $\mathbf{C}$:

$$f(\boldsymbol{y}) \leq f(\boldsymbol{x}) + \langle \nabla f(\boldsymbol{x}), \boldsymbol{y} - \boldsymbol{x} \rangle + \frac{L}{2}\|\boldsymbol{y} - \boldsymbol{x}\|^2 = \boxed{f(\boldsymbol{x}) + 2\langle \boldsymbol{x} - \mathcal{P}_{\mathbf{C}}(\boldsymbol{x}), \boldsymbol{y} - \boldsymbol{x}\rangle + \frac{1}{\beta}\|\boldsymbol{y} - \boldsymbol{x}\|^2}$$

Applying this lemma, let us use $\boldsymbol{x} = \boldsymbol{x}'_{t-1}$ and $\boldsymbol{y} = \boldsymbol{x}'_t$. Noting that $\mathcal{P}_{\mathbf{C}}(\boldsymbol{x}'_t) = \boldsymbol{x}_{t-1}$, we get:

$$\mathrm{dist}(\boldsymbol{x}'_t, \mathbf{C})^2 \leq \underbrace{\mathrm{dist}(\boldsymbol{x}'_{t-1}, \mathbf{C})^2}_{\text{Term A}} + \underbrace{2\,\langle \boldsymbol{x}'_{t-1} - \boldsymbol{x}_{t-2}, \boldsymbol{x}'_t - \boldsymbol{x}'_{t-1}\rangle}_{\text{Term B}} + \underbrace{\frac{1}{\beta}\|\boldsymbol{x}'_t - \boldsymbol{x}'_{t-1}\|^2}_{\text{Term C}} \qquad (\star)$$

**Decomposing Term B.**   First, consider that since the step size is decreasing $\gamma_t \geq \gamma_{t-1}$:

$$\boldsymbol{x}'_{t-1} - \boldsymbol{x}_{t-2} \leq (\boldsymbol{x}_{t-1} - \boldsymbol{x}_{t-2}) + \gamma_{t-1}s_{t-1}$$
$$\leq (\boldsymbol{x}_{t-1} - \boldsymbol{x}_{t-2}) + \gamma_t s_{t-1}$$

By the same rationale,

$$\boldsymbol{x}'_t - \boldsymbol{x}'_{t-1} \leq (\boldsymbol{x}_t - \boldsymbol{x}_{t-1}) + \gamma_t(s_t - s_{t-1}). \qquad \text{(Definition B.1)}$$

*Proof of non-expansiveness of the projection operator.*   Next, we prove the non-expansiveness of the projection operator:

$$\|\boldsymbol{x}_t - \boldsymbol{x}_{t-1}\| \leq 2\,\gamma_{t+1}G^2 \qquad (\mathcal{L}^+)$$

Given $\boldsymbol{x}_t = \mathcal{P}_{\mathbf{C}}(\boldsymbol{x}'_{t+1})$ and $\boldsymbol{x}_{t-1} = \mathcal{P}_{\mathbf{C}}(\boldsymbol{x}'_t)$,

$$\|\boldsymbol{x}_t - \boldsymbol{x}_{t-1}\| = \|\mathcal{P}_{\mathbf{C}}(\boldsymbol{x}'_{t+1}) - \mathcal{P}_{\mathbf{C}}(\boldsymbol{x}'_t)\| \leq \|\boldsymbol{x}_{t+1} - \boldsymbol{x}_t\|$$

since projections onto closed prox-regular sets are $L$-Lipshitz.

Now:

$$\boldsymbol{x}'_{t+1} = \boldsymbol{x}_{t+1} + \gamma_{t+1}s_{t+1};$$
$$\boldsymbol{x}'_t = \boldsymbol{x}_t + \gamma_t s_t;$$
$$\boldsymbol{x}'_{t+1} - \boldsymbol{x}'_t = (\boldsymbol{x}_{t+1} - \boldsymbol{x}_t) + (\gamma_{t+1}s_{t+1} - \gamma_t s_t). \qquad \text{(Definition B.2)}$$

Making the projection residual,

$$\boldsymbol{x}_{t+1} - \boldsymbol{x}_t = \boldsymbol{x}_{t+1} - \mathcal{P}_{\mathbf{C}}(\boldsymbol{x}'_{t+1})$$

orthogonal to the target space at $\boldsymbol{x}_t$ (and any vector of the form $s_{t+1} - s_t$). Thus, since $\|s_t\| \leq G \ \ \forall t$:

$$\|\boldsymbol{x}'_{t+1} - \boldsymbol{x}'_t\|^2 = \|\gamma_{t+1}s_{t+1}\|^2 + \|\gamma_t s_t\| \leq (\gamma_{t+1}^2 + \gamma_t^2)G^2$$

Taking the square root:

$$\|\boldsymbol{x}'_{t+1} - \boldsymbol{x}'_t\| \leq \sqrt{\gamma_{t+1}^2 + \gamma_t^2}\,G$$

Since $\gamma_{t+1} \geq \gamma_t$:

$$\|\boldsymbol{x}'_{t+1} - \boldsymbol{x}'_t\| \leq \sqrt{2}\gamma_{t+1}G$$
$$< 2\gamma_{t+1}G$$

Finally, by applying Definition (B.1), $\|\boldsymbol{x}_{t+1} - \boldsymbol{x}_t\| \leq \|\boldsymbol{x}'_{t+1} - \boldsymbol{x}'_t\|$, and thus:

$$\boxed{\|\boldsymbol{x}_{t+1} - \boldsymbol{x}_t\| \leq 2\gamma_{t+1}G}$$

$\square$

Now, prox-regularity gives:

$$\langle \boldsymbol{x}'_{t-1} - \boldsymbol{x}_{t-2}, \boldsymbol{x}'_t - \boldsymbol{x}'_{t-1}\rangle \leq \beta\|\boldsymbol{x}_t - \boldsymbol{x}_{t-1}\|^2$$
$$\leq 4\beta\gamma_{t+1}^2 G^2 \qquad \text{(Bound B.1)}$$

where the Bound B.1 is derived by applying $(\mathcal{L}^+)$.

Since $\mathbf{C}$ in $\beta$-prox regular, for any point $u$ near $\mathbf{C}$ and $v \in \mathbf{C}$:

$$\langle u - \mathcal{P}_{\mathbf{C}}(u), v - \mathcal{P}_{\mathbf{C}}(u) \rangle \leq \beta \|v - \mathcal{P}_{\mathbf{C}}(u)\|^2$$

Above, we substitute:

$$u = \boldsymbol{x}'_t = \boldsymbol{x}_t + \gamma_t s_t$$
$$v = \boldsymbol{x}_t$$
$$\mathcal{P}_{\mathbf{C}}(u) = \boldsymbol{x}_{t-1}$$

Now, expanding the inner product:

$$\begin{aligned}
\langle \boldsymbol{x}'_{t-1} - \boldsymbol{x}_{t-2}, \boldsymbol{x}'_t - \boldsymbol{x}'_{t-1} \rangle &= \langle \boldsymbol{x}'_{t-1} - \boldsymbol{x}_{t-2}, (\boldsymbol{x}_t + \gamma_t s_t) - (\boldsymbol{x}_{t-1} + \gamma_{t-1} s_{t-1}) \rangle \\
&\leq \langle \boldsymbol{x}'_{t-1} - \boldsymbol{x}_{t-2}, (\boldsymbol{x}_t - \boldsymbol{x}_{t-1}) + \gamma_t (s_t - s_{t-1}) \rangle \\
&\leq \langle \boldsymbol{x}'_{t-1} - \boldsymbol{x}_{t-2}, (\boldsymbol{x}_t - \boldsymbol{x}_{t-1}) \rangle + \langle \boldsymbol{x}'_{t-1} - \boldsymbol{x}_{t-2}, \gamma_t (s_t - s_{t-1}) \rangle
\end{aligned}$$

and since $\|s_t\| \leq G \quad \forall t$: $\langle s_{t+1}, s_t \rangle \leq \|s_{t+1}\| \|s_t\| \leq G^2$ so $\langle s_{t-1}, s_t \rangle - \|s_{t+1}\|^2 \leq G^2$, and:

$$\langle \boldsymbol{x}'_{t-1} - \boldsymbol{x}_{t-2}, \gamma_t (s_t - s_{t-1}) \rangle \leq \gamma_t^2 G^2 \tag{Bound B.2}$$

By applying Definition (B.2):

$$\begin{aligned}
\langle \boldsymbol{x}'_{t-1} - \boldsymbol{x}_{t-2}, \boldsymbol{x}'_t - \boldsymbol{x}'_{t-1} \rangle &= \langle \boldsymbol{x}'_{t-1} - \boldsymbol{x}_{t-2}, (\boldsymbol{x}_t - \boldsymbol{x}_t) + (\gamma_t s_t - \gamma_{t-1} s_{t-1}) \rangle \\
&\leq \langle \boldsymbol{x}'_{t-1} - \boldsymbol{x}_{t-2}, (\boldsymbol{x}_t - \boldsymbol{x}_{t-1}) \rangle
\end{aligned}$$

Therefore, by applying Bound (B.1) to the previous inequality and Bound (B.2) directly, Term B is upper bounded by:

$$\boxed{2 \langle \boldsymbol{x}'_{t-1} - \boldsymbol{x}_{t-2}, \boldsymbol{x}'_t - \boldsymbol{x}'_{t-1} \rangle \leq 8\beta \gamma_{t+1}^2 G^2 + 2\gamma_t^2 G^2} \tag{Bound B.3}$$

**Decomposing Term C.** Next, we derive a bound on Term C in Eq. ($\star$). As already shown,

$$\|\boldsymbol{x}'_t - \boldsymbol{x}'_{t-1}\| \leq 4\gamma_t G,$$

given:

$$\begin{aligned}
\boldsymbol{x}'_t - \boldsymbol{x}'_{t-1} &\leq \underbrace{(\boldsymbol{x}_t - \boldsymbol{x}_{t-1})}_{\leq 2\gamma_{t+1} G} + \underbrace{\gamma_t (s_t - s_{t-1})}_{\leq 2G} \\
&\leq 4\gamma_{t+1} G
\end{aligned}$$

Thus,

$$\boxed{\frac{1}{\beta} \|\boldsymbol{x}'_t - \boldsymbol{x}'_{t-1}\|^2 \leq \frac{16}{\beta} \gamma_{t+1}^2 G^2} \tag{Bound C.1}$$

Combining bounds (B.3) and (C.1) into ($\star$), and recalling that $\gamma_{t+1} \geq \gamma_t$:

$$\text{dist}(\boldsymbol{x}'_t, \mathbf{C})^2 \leq \underbrace{\text{dist}(\boldsymbol{x}'_{t-1}, \mathbf{C})^2}_{d} + \underbrace{(8\beta + 2 + \frac{16}{\beta}) \gamma_{t+1}^2 G^2}_{K}$$

Now, we rewrite Term A, which for ease of notation we will refer to as $d$:

$$d^2 = (1 - 2\beta\gamma_{t+1})d^2 + 2\beta\gamma_{t+1}d^2$$

Thus:

$$\begin{aligned}
\text{dist}(\boldsymbol{x}'_t, \mathbf{C})^2 &\leq d^2 - 2\beta\gamma_{t+1}d^2 + 2\beta\gamma_{t+1}d^2 + K\gamma_{t+1}^2 G^2 \\
&= (1 - 2\beta\gamma_{t+1})d^2 + \left[ 2\beta\gamma_{t+1}d^2 + K\gamma_{t+1}^2 G^2 \right]
\end{aligned}$$

Next, through Young's inequality, we simplify this expression further.

**Theorem H.2. (Young's Inequality)** $\forall u, v \geq 0, \epsilon > 0$:

$$uv \leq \frac{u^2}{2\epsilon} + \frac{\epsilon v^2}{2}$$

If we choose $u = \sqrt{2\beta\gamma_{t+1}}d$, $v = \sqrt{K}\gamma_{t+1}G$, and $\epsilon = \frac{2\beta d^2}{k\gamma_{t+1}G^2}$, then

$$uv = \sqrt{2\beta\gamma_{t+1}}d \times \sqrt{K}\gamma_{t+1}G$$
$$= \sqrt{2K}\gamma_{t+1}^{\frac{3}{4}}Gd$$

Applying Young's Inequality:

$$uv \leq \frac{u^2}{2\epsilon} + \frac{\epsilon v^2}{2}$$
$$= \frac{2\beta\gamma_{t+1}d^2}{2(\frac{2\beta d^2}{K\gamma_{t+1}G^2})} + \frac{\epsilon v^2}{2}$$
$$= \frac{K\gamma_{t+1}^2 G^2}{2} + \frac{\epsilon v^2}{2}$$
$$= \frac{K\gamma_{t+1}^2 G^2}{2} + \left(\frac{1}{2} \times \frac{2\beta d^2}{K\gamma_{t+1}G^2} \times K\gamma_{t+1}^2 G^2\right)$$
$$= \frac{K\gamma_{t+1}^2 G^2}{2} + \beta\gamma_{t+1}d^2$$

Thus,

$$\sqrt{2K}\gamma_{t+1}^{\frac{3}{4}}Gd \leq \frac{K\gamma_{t+1}^2 G^2}{2} + \beta\gamma_{t+1}d^2$$

Finally, taken altogether:

$$2\beta\gamma_{t+1}d^2 + K\gamma_{t+1}^2 G^2 \leq \beta\gamma_{t+1}d^2 + \left(\frac{K\gamma_{t+1}^2 G^2}{2} + \beta\gamma_{t+1}d^2\right)$$
$$= 2\beta\gamma_{t+1}d^2 + \frac{K}{2}\gamma_{t+1}^2 G^2$$

Since $\gamma_{t+1} \leq \frac{\beta}{2G^2}$, then

$$\frac{K}{2}\gamma_{t+1}^2 G^2 \leq \frac{1}{2}\left(8\beta + 2 + \frac{16}{\beta}\right)\frac{\beta^2}{4G^2} = \mathcal{O}(\beta^3)$$

which is bounded by $\gamma_{t+1}^2 G^2$ for all $\beta \geq 0$.

Thus,

$$2\beta\gamma_{t+1}d^2 + K\gamma_{t+1}^2 G^2 \leq 2\beta\gamma_{t+1}d^2 + \gamma_{t+1}^2 G^2.$$

Finally, by substitution:

$$\boxed{\text{dist}(\boldsymbol{x}_t', \mathbf{C})^2 \leq (1 - 2\beta\gamma_{t+1})\text{dist}(\boldsymbol{x}_{t-1}', \mathbf{C})^2 + \underbrace{\gamma_{t+1}^2 G^2}_{\mathcal{O}(\beta^3)}}$$

$\square$

## H.2 Proof of Theorem 4.1 (Part 2): Non-Asymptotic Feasibility in the Latent Space

*Proof.* The previous proof can be naturally extended to latent space models under smoothness assumptions:

**Assumption H.3.**

(1) **(Decoder regularity)** $\mathcal{D} : \mathbb{R}^{d_z} \to \mathbb{R}^{d_x}$ is $\ell$-Lipschitz and $\|\nabla\mathcal{D}(\mathbf{z})\|_{\text{op}} \leq \ell$ for all $\mathbf{z}$.

(2) **(Constraint regularity)** $g : \mathbb{R}^{d_x} \to \mathbb{R}^m$ is $L$-smooth and $\nabla g(\mathbf{x})$ has full row rank on $\mathbf{C} = \{\mathbf{x} : g(\mathbf{x}) = \mathbf{0}\}$, hence $\mathbf{C}$ is prox-regular.

Let $\mathbf{x}_t' = \mathcal{D}(\mathbf{z}_t')$ be the decoded pre-projection iterate. Prox-regularity implies

$$\left\|\mathbf{x}_{t+1} - \mathcal{P}_{\mathbf{C}}(\mathbf{x}_t')\right\| \leq (1 - \beta)\left\|\mathbf{x}_t' - \mathcal{P}_{\mathbf{C}}(\mathbf{x}_t')\right\| = (1 - \beta)\,\text{dist}(\mathbf{x}_t', \mathbf{C}),$$

where $\mathbf{x}_{t+1} = \mathcal{P}_{\mathbf{C}}(\mathbf{x}_t')$ is the data-space projection produced by the algorithm.

Because $\mathcal{D}$ is $\ell$-Lipschitz we have

$$\text{dist}(\mathbf{x}_t', \mathbf{C}) = \text{dist}\big(\mathcal{D}(\mathbf{z}_t'), \mathbf{C}\big) \leq \ell\,\text{dist}\big(\mathbf{z}_t', \mathcal{D}^{-1}(\mathbf{C})\big),$$

where the inverse image is $\mathcal{D}^{-1}(\mathbf{C}) = \{\mathbf{z} : g(\mathcal{D}(\mathbf{z})) = \mathbf{0}\}$. Using the same prox-regularity argument (now applied to the level set of $g \circ \mathcal{D}$), Thus, by the chain-rule and prox-regularity:

$$\|\nabla(g \circ \mathcal{D})(\mathbf{z}) - \nabla(g \circ \mathcal{D})(\mathbf{z}')\| \le \ell L \|\mathbf{z} - \mathbf{z}'\|,$$

which shows that for a projection in latent space, the contraction factor $(1 - \beta)$ is replaced with

$$1 - \beta' := 1 - \beta/(\ell L).$$

Therefore, denoting by $\mathbf{z}'_{t+1}$ the latent iterate after one full update but *before* decoding, we obtain the fundamental recursion

$$\boxed{\operatorname{dist}\!\big(\mathcal{D}(\mathbf{z}'_t), \mathbf{C}\big)^2 \;\le\; (1 - 2\beta'\gamma_{t+1})\operatorname{dist}\!\big(\mathcal{D}(\mathbf{z}'_{t+1}), \mathbf{C}\big)^2 + \gamma_{t+1}^2 G^2,} \qquad \text{(non-asymptotic feasibility)}$$

where $G$ is the same (finite) uniform bound used in Part 1. $\qquad\square$

## H.3 Proof of Theorem 4.2: Fidelity

*Proof.* Finally, we prove the fidelity bound:

$$\boxed{\operatorname{KL}\!\big(q(\mathbf{z}_{t-1}) \,\|\, p_{\text{data}}\big) \le \operatorname{KL}\!\big(q(\mathbf{z}_t) \| p_{\text{data}}\big) + \gamma_t G^2}$$

We begin by assuming feasibility of the training set $p_{\text{data}} \subseteq \mathbf{C}$. Provided this holds, but the data-processing inequality,

$$\begin{aligned}
\operatorname{KL}\!\big(q(\mathbf{z}_{t-1}) \,\|\, p_{\text{data}}\big) &= \operatorname{KL}\!\big(\mathcal{P}_{\mathbf{C}}(q(\mathbf{z}'_t)) \,\|\, \mathcal{P}_{\mathbf{C}}(p_{\text{data}})\big) \\
&= \operatorname{KL}\!\big(\mathcal{P}_{\mathbf{C}}(q(\mathbf{z}'_t)) \,\|\, p_{\text{data}}\big) \\
&\le \operatorname{KL}\!\big(q(\mathbf{z}'_t) \,\|\, p_{\text{data}}\big)
\end{aligned}$$

This result shows that since the projection operator is a deterministic, measurable mapping, the KL will not be increased by this operation *iff* the training set is a subset of the constraint set (all samples in the distribution fall in $\mathbf{C}$). Hence, subsequently we will ignore the presence of this operator in obtaining our bound.

Next, consider that while in continuous time the application of the true score results in a strictly monotonically decreasing KL, Euler discretizations introduce a bounded error.

Consider that $\mathbf{z}'_t = \mathbf{z}_t + \gamma_t s_t$. Each denoising step shifts the vector $\mathbf{z}_t$ by a vector norm of at most $\gamma_t G$, provided that $\|s_t\| \le G$.

**Lemma H.4.** *Suppose $\|s_t\| \le G$ everywhere, Then,*

$$\operatorname{KL}\!\big(q(\mathbf{z}'_t) \,\|\, p_{\text{data}}\big) - \operatorname{KL}\!\big(q(\mathbf{z}_t) \,\|\, p_{\text{data}}\big) \le \gamma_t G^2.$$

*Proof of Lemma H.4.*

$$\operatorname{KL}\!\big(q(\mathbf{z}_t) \,\|\, p_{\text{data}}\big) = \int q(\mathbf{z}_t) \log \frac{q(\mathbf{z}_t)}{p_{\text{data}}} dx,$$

and

$$\operatorname{KL}\!\big(q(\mathbf{z}'_t) \,\|\, p_{\text{data}}\big) = \int q(\mathbf{z}'_t) \log \frac{q(\mathbf{z}'_t)}{p_{\text{data}}} dx,$$

Therefore,

$$\operatorname{KL}\!\big(q(\mathbf{z}'_t) \,\|\, p_{\text{data}}\big) - \operatorname{KL}\!\big(q(\mathbf{z}_t) \,\|\, p_{\text{data}}\big) = \int q(\mathbf{z}_t) \left[\log q(\mathbf{z}_t) - \log q(\mathbf{z}_t + \gamma_t s_t)\right] dx,$$

Since $\|s_t\| \le G$, we assume that the true score is bounded by a comparable norm $\|\nabla \log p(\mathbf{z}_t)\| \le G$. A one-step Taylor bound gives:

$$\log p(\mathbf{z}_t) - \log p(\mathbf{z}_t + \gamma_t s_t) \le \|\gamma_t s_t\| \cdot \|\nabla \log p(\xi)\| \le \gamma_t G^2$$

for some $\xi$ on the line segment between $\mathbf{z}_t$ and $\mathbf{z}'_t$. Integrating over $q(\mathbf{z}_t)$ gives

$$\boxed{\operatorname{KL}\!\big(q(\mathbf{z}'_t) \,\|\, p_{\text{data}}\big) - \operatorname{KL}\!\big(q(\mathbf{z}_t) \,\|\, p_{\text{data}}\big) \le \gamma_t G^2}$$

$\qquad\square$

Lemma H.4 can then be directly applied to derive our bound:

$$\boxed{\operatorname{KL}\!\big(q(\mathbf{z}_{t-1}) \,\|\, p_{\text{data}}\big) \le \operatorname{KL}\!\big(q(\mathbf{z}_t) \| p_{\text{data}}\big) + \gamma_t G^2}$$

$\qquad\square$

