# OpenReview forum: "Training-Free Constrained Generation With Stable Diffusion Models"
_NeurIPS.cc/2025/Conference — NeurIPS 2025 spotlight_

### Official Review · Reviewer_1Dzz · 2025-07-02

**Clarity:** 4
**Significance:** 4
**Originality:** 4
**Rating:** 5
**Confidence:** 5

**Summary:**

This paper presents a framework that integrates constrained optimization with stable diffusion models to enable constraint-aware data generation.  The proposed approach introduces a constraint formulation that can be incorporated into the diffusion sampling process, allowing for the generation of outputs that satisfy stringent physical, functional, and legal requirements while maintaining high synthesis quality. Two theorems provides convergence properties in theory. Through experiments, the authors demonstrate the framework's ability to handle complex constraint in stable diffusion models.

**Questions:**

1. Is the speed of the methods depending on the complexity of the constraint?
2. How to evluate the complex of the constraint? Is there some numerical value to measure that？

**Ethical Concerns:**

["NO or VERY MINOR ethics concerns only"]

**Limitations:**

Yes.

**Paper Formatting Concerns:**

The format is good.

**Quality:**

4

**Strengths And Weaknesses:**

The strengths are introducing a frameworks with constrained optimization in stable diffusion models.   The detailed proof of their convergence represent a non-trivial and significant research endeavor with both theoretical and practical implications.

---

> ### Author Rebuttal · Authors · 2025-07-30
>
> Thank you for the time invested in reviewing our work and for praising the innovative aspects of the proposed method. We will address the point raised below and will integrate the suggestions in our final version.
>
> > **Q1:** Is the speed of the methods depending on the complexity of the constraint?
>
> Yes. The performance of the proposed method depends largely on the complexity of the constraints enforced and the ease with which their violations can be measured. In the paper, *we presented three cases with varying levels of complexity (and execution times)* and reported runtimes in Appendix F.
>
>
> > **Q2:** How to evaluate the complexity of the constraint? Is there some numerical value to measure that？
>
> Generally, one may want to categorize them in convex and non-convex, but in this paper, we go well beyond non-convexity as we include non-differentiable black box simulators that are used to constrain our model. To provide some details:
>
> - **Section 6.1:** This is a knapsack constraint. While in general knapsack problems are NP-complete, the version adopted (which uses integers as weight) is known to be weakly NP-complete and admits a fully polynomial approximation. It is solved efficiently in $O(nm)$ where $n$ is the number of pixels and $m$ is the number of values to compute for the dynamic program.
>
> - **Sections 6.2 and 6.3:** The inclusion of a blackbox simulator and a nonconvex neural surrogate characterizes these problems as NP-complete. This makes these settings complex but also very compelling.
>
> Overall, it is evident that evaluating a constraint through a simple differentiable operation on the sample (e.g., microstructure porosity) is straightforward and leads to a fast and efficient correction algorithm. In contrast, evaluating a constraint using external numerical simulators or, more generally, black-box models (e.g., as in the case of our metamaterials experiment) involves higher computational costs and greater implementation complexity.
>
> ---
>
> Thank you again for your review and your praise for our work. We believe that our responses have addressed each concern presented in your assessment but we are more than happy to provide additional details as needed. Many thanks!

---

> > ### Author Response · Authors · 2025-08-07
> >
> > As the discussion phase will conclude in about two days, we would appreciate your feedback to ensure that any further questions or revisions can be addressed in a timely manner. If our responses have satisfactorily addressed your concerns, let us take this opportunity to thank you for your valuable input.
> >
> > We appreciate your time and consideration of our work!

---

### Official Review · Reviewer_2eXJ · 2025-07-02

**Clarity:** 4
**Significance:** 4
**Originality:** 3
**Rating:** 6
**Confidence:** 2

**Summary:**

The paper presents a novel approach for constraining latent diffusion models, such as Stable Diffusion, based on constraints formulated in the original (e.g., image) space. Then, the constraints are projected via the decoder into the latent space of the diffusion model and subsequently enforced during each diffusion step via an inner optimization loop akin to projected gradient descent. It is important to stress that this is done after training during inference and the gradients with respect to the model parameters are not optimized. The constraints can even be possibly non-convex and not differentiable. To remedy situations where the constraints are not differentiable out of the box (e.g., where the constraint is the outcome of an experiment or simulation), the authors adopt existing approaches from literature, such as differentiable surrogates or smoothed approximated gradients via random local perturbations, to estimate the gradients of the constraints.
The method is quantitatively evaluated on three different domains: microstructure generation, metamaterial inverse design, and copyright-safe generation and compared against a text-conditioned diffusion model and a baseline method that enforces the constraints postpriori directly in the output/image space.

**Questions:**

- How would training-time enforcement of the constraints compare to the proposed approach? Would it make sense to adopt a baseline approach that trains a model with the constraints enforced during training, and then compare the performance of the proposed method to that baseline? If not, why not?
- For example in Equation (6), the authors introduce the trade-off between constraint satisfaction and maintaining similarity to the original learned distribution. Could instead of this soft constraint also a hard constraint be used? If so, how would this affect the performance of the method?
- In line 237, the authors mention a smoothing function that is used. Where is the smoothing function defined? Is it just the sum operator?
- Figure 8: Does the performance get even better when you perform more iterations/steps? Why did you stop at four steps?

**Ethical Concerns:**

["NO or VERY MINOR ethics concerns only"]

**Final Justification:**

The reviewer appreciates the author's rebuttal and finds that it addresses the reviewer's questions and concerns to satisfaction. The reviewer keeps the score of "6: Strong Accept" after studying the rebuttal and the reviews and rebuttal to other reviews.

- Indeed, the small number of baselines is also highlighted by other reviewers, but the reviewer is not sufficiently familiar with the literature in this domain to assess with confidence if there would be another suitable baseline that would have needed to be included or not.
- The paper is already well written and promises to be even more improved in quality after the authors have implemented the promised changes and clarifications.

**Limitations:**

Although the paper does discuss limitations in Appendix A, the reviewer would like to see a few-sentence summary of the limitations included in the main text of the paper (e.g., in the conclusion section).

**Paper Formatting Concerns:**

The reviewer did not identify any paper formatting concerns.

**Quality:**

4

**Strengths And Weaknesses:**

## Strengths

- The paper is very well written and clearly structured.
- The paper tackles a very timely and relevant problem in the field of generative models. The ability to infuse physical priors, constraints, or other guarantees into the generation process without sacrificing the quality of the generated samples (too much) is a very important capability (e.g., with respect to video-generation models) and this paper could have a significant impact on this domain.
- The method is evaluated on problem settings from three different domains and the quantitative results are strong.

## Weaknesses

- Some of the figures are a bit unclear - also, because the captions are very brief and do not explain the figures in detail. Instead, a reader should be able to understand the figures (at least their gist) without having to read the main text. Examples include Figure 4 and Figure 5.
- Although the two considered baseline methods are suitable and well-chosen, it could be beneficial to include additional baselines from related work, or, for example, a baseline that enforces the constraints during training. This would allow for a more comprehensive evaluation of the proposed method.
- The computational complexity, efficiency and runtime of the method is not discuss in detail.

### Detailed Comments on Minor Issues

- Line 26: What is the difference between "physical laws" and "first principles"? Are physical laws not a subset of first principles? Please clarify.
- Line 36: Although the authors later define how stable diffusion models operate in a learned latent space, at this point in the text it might not clear to a reader who is not familiar with latent diffusion models what a "latent diffusion model" is. One subsentence could be added to clarify this.
- Figure 4: To the reviewer, it is not clear what is the meaning of the different rows in the "Structural analysis" section of the figure. Please define in the caption.
- "Differentiating through blackbox simulators" lines 243-244: It seems a bit strange that there is no division of the equation by the (direction) of perturbation or something similar needed. Why is the case? Usually, you would have in finite differences something of the form $\frac{f(x + \epsilon) - f(x)}{\epsilon}$ with $\epsilon \in \mathbb{R}$, but here it seems like the authors are just using $f(x + \epsilon) - f(x)$.
- The heading of Appendix H "Missing Proofs" sounds a bit informal or even confusing. A more formal heading could be "Supplementary Proofs" or something similar.

---

> ### Author Rebuttal · Authors · 2025-07-30
>
> Thank you for the time you dedicated to this detailed and constructive review! We appreciate that the strengths of our work were identified and highlighted, and we find all the comments helpful for further improving it. We address each of the points in detail below.
>
> > **W1 & C3:** Some of the figures are a bit unclear - also, because the captions are very brief and do not explain the figures in detail.
>
> Thank you for your comment. We agree this as an area for improvement and will revise the captions accordingly:
>
> - **Figure 4:** Successive steps of DPO. The sample is iteratively improved and the stress-strain curve aligns with the target.
> Structural analysis shows the progressive deformation under controlled compression.
>
> - **Figure 5:** Left: Denoising process of Cond vs. Latent (Ours). Out method drives the denoising toward a copyright-safe image. Top-right: showing projection from original (O) to projected (P) in the PCA-2 space. Bottom-right: Constraint satisfaction and FID scores.
>
>
> > **W2:** It could be beneficial to include additional baselines from related work, or, for example, a baseline that enforces the constraints during training.
>
> During development we indeed evaluated several additional baselines (which are discussed in Appendix E) but, due to space constraints, *we included those that either represent the state-of-the-art in specific domains* (Christopher et al. (NeurIPS 2024) and  Bastek & Kochmann (Nature 2023) for microstructure generation and metamaterial design, respectively) or that we considered to be *the best performing among existing alternatives* (text-conditional model for copyright-safe generation).
>
> We also highlight that in the cases of metamaterials and copyright, *the selected baselines indeed incorporate constraints during the training phase*. For example, Bastek & Kochmann impose periodic structure constraints during training. Similarly, constraints are encoded in Section 6.3 using a text-conditional model, which closely resembles existing work for training-time latent diffusion methods for constrained inverse-design settings [1]. Additionally, given the complexity of the constraints modeled in this paper (e.g., in Sections 6.2 we use a finite element analysis solver is used in the loop, and in Section 6.3 the constraints are highly nonconvex) very few existing methods are applicable. To the best of our knowledge, *we compare to the strongest methods that currently exist for the settings explored*.
> We will, however, empahsize in the paper, the additional baselines discussed in Appendix E.
>
>
> > **W3:** The computational complexity, efficiency and runtime of the method is not discuss in detail.
>
> We are happy to expand on this point. Appendix F provides the requested runtimes, but we will make sure to give them greater visibility in the final version of the paper. In terms of constraint complexity:
>
> - **Section 6.1:** This is a knapsack constraint. While in general knapsack problems are NP-complete, the version adopted (which uses integers as weight) is known to be weakly NP-complete and admits a fully polynomial approximation. It is solved efficiently in $O(nm)$ where $n$ is the number of pixels and $m$ is the number of values to compute for the dynamic program.
>
> - **Sections 6.2 and 6.3:** The inclusion of a blackbox simulator and a nonconvex neural surrogate characterizes these problems as NP-complete, but also quite compelling applications.
>
> We will report these details in the final version of the paper. Thanks for the suggestion.
>
>
> > **C1:**  What is the difference between "physical laws" and "first principles"?
>
> First principles are fundamental concepts; physical laws are their mathematical consequences or empirical summaries. We favor the law form when writing constraints. In practice, our usage partially overlaps; we will clarify as needed.
>
>
> > **C2:** Although the authors later define how stable diffusion models operate in a learned latent space, at this point in the text it might not clear to a reader who is not familiar with latent diffusion models what a "latent diffusion model" is.
>
> This is a good observation, thank you. We will clarify.
>
>
> > **C4:** It seems a bit strange that there is no division of the equation by the (direction) of perturbation or something similar needed. Why is the case?
>
> We will add a short derivation to show that, for the chosen step size, the direction scaling is absorbed into the proximal update, eliminating an explicit division.
>
>
> > **C5:** A more formal heading could be "Supplementary Proofs" or something similar.
>
> Agreed. Thank you!
>
>
> > **Q1:** How would training-time enforcement of the constraints compare to the proposed approach
>
> Thank you for the question, since this is one of our contributions. Besides requiring heavier training, we see two key limitations of training-time enforcement methods: (1) they cannot guarantee constraint satisfaction even for the case of convex constraints, and (2) It does not support generalization to unseen constraints, since the model only learns to ``satisfy'' (we quote the word satisfy here because this is only true at a distribution-level, not at the level of each sample) those present during training. These are two key strengths of our proposed approach.
> Interestingly, we also find that this allows us to fine-tune models using very sparse datasets, a property we do not observe with training-time methods. This raises a broader and open question: _why does sampling-time enforcement enable generalization under limited supervision?_ We are actively investigating this.
>
> Additionally, let us point out that some of the baselines we compare against do use training-time constraint enforcement, so the empirical comparisons reflect this distinction.
>
>
> > **Q2:** For example in Equation (6), the authors introduce the trade-off between constraint satisfaction and maintaining similarity to the original learned distribution. Could instead of this soft constraint also a hard constraint be used?
>
> Yes, in principle, and we do so when the constraint is treated as an indicator function. However, the hard version requires solving a projection exactly at every step and requires a closed form solution. This is impractical for our black‑box constraints settings (Sections 6.2 and 6.3). The soft penalty gives a practical balance.
>
>
> > **Q3:** Where is the smoothing function defined? Is it just the sum operator?
>
> Thanks for this question, since this is a key contribution of our work but, we agree its description would deserve additional space to make full justice to it.
>
> The *smoothing function* is the map
>
> $$
> \Phi_\nu(\mathbf{x}) = \mathbb{E}_{\varepsilon} \left[ \phi(\mathbf{x} + \nu\varepsilon)\right],
> $$
>
> where $\phi$ is the non‑differentiable simulator, $\varepsilon \sim \mathcal{N}(0,I)$ (or any full‑support noise, really) and $\nu>0$ is a temperature that scales the perturbation.  Averaging the simulator's output over these random local perturbations _smooths away the simulator’s discontinuities_: for any fixed $\nu$, $\Phi_\nu$ is continuously differentiable even though $\phi$ may not be.
> As $\nu \to 0$ the bias vanishes and $\Phi_\nu$ approaches the raw simulator; as $\nu$ grows, the function becomes smoother but drifts toward the mean behavior of the simulator in a neighborhood of $\mathbf{x}$.
>
> To make this practical, we estimate the expectation with $M$ Monte‑Carlo samples,
> $$
> \bar{\phi}_\epsilon(\mathrm{x}) = \frac{1}{M} \left[ \phi(\mathrm{x} + \nu \epsilon^{(1)}) + \cdots + \phi(\mathrm{x} + \nu \epsilon^{(M)}) \right]
> $$
> and treat the sample average as a differentiable proxy.
>
> Now, because the noise is injected *outside* the simulator, gradients flow through the re‑parameterization $\nabla_{\mathbf{x}} \Phi_\nu(\mathbf{x}) = \frac{1}{\nu}
> \mathbb{E}\left[ \phi(\mathbf{x} + \nu\varepsilon) \varepsilon\right]$.
> The same formula with the finite‑sample average gives an unbiased gradient estimator that we use into our proximal Langevin update.
>
> We will make sure to extend this discussion.
>
>
> > **Q4:** Figure 8: Does the performance get even better when you perform more iterations/steps?
>
> Yes. By proceeding with more steps, it is in principle possible to achieve even smaller errors, as we have empirically observed. Since the computation time increases, however, it is also important to strike a good trade-off between these two desiderata.
>
>
> ---
>
> Thank you again for your review and for championing our work. We believe our response has clarified all the outstanding questions, but we are happy to provide additional details as needed.
>
>
>
> [1] Dontas, Michail, et al. "Blind Inverse Problem Solving Made Easy by Text-to-Image Latent Diffusion." arXiv preprint arXiv:2412.00557 (2024).

---

> > ### Comment · Reviewer_2eXJ · 2025-08-02
> > **Response to Rebuttal and a few more Questions**
> >
> > The reviewer thanks the authors for their detailed and constructive response to the initial review. The clarifications provided address most of the concerns raised in the review, and the reviewer appreciates the authors' willingness to improve the paper based on the feedback. The reviewer is happy with the rebuttal and believes it addressed the reviewer's concerns. A few small questions and comments remain and are listed below.
> >
> > > Successive steps of DPO. The sample is iteratively improved and the stress-strain curve aligns with the target. Structural analysis shows the progressive deformation under controlled compression.
> >
> > There should also be an explanation about the meaning of the different rows in the "Structural analysis" section of Figure 4. The caption should clarify this.
> >
> > > First principles are fundamental concepts; physical laws are their mathematical consequences or empirical summaries. We favor the law form when writing constraints. In practice, our usage partially overlaps; we will clarify as needed.
> >
> > The reviewer thanks the authors for clarifying this point. I would say that most (non-empirical) physical models/laws are derived from first principles, but this might be a matter of terminology and a subjective interpretation. The reviewer is happy with the clarification that the authors will provide in the paper.
> >
> > > We will add a short derivation to show that, for the chosen step size, the direction scaling is absorbed into the proximal update, eliminating an explicit division.
> >
> > Can you intuitively explain here (and later in the paper) why this is the case?
> >
> > > Thank you for the question, since this is one of our contributions. Besides requiring heavier training, we see two key limitations of training-time enforcement methods: (1) they cannot guarantee constraint satisfaction even for the case of convex constraints, and (2) It does not support generalization to unseen constraints, since the model only learns to ``satisfy'' (we quote the word satisfy here because this is only true at a distribution-level, not at the level of each sample) those present during training. These are two key strengths of our proposed approach. Interestingly, we also find that this allows us to fine-tune models using very sparse datasets, a property we do not observe with training-time methods. This raises a broader and open question: why does sampling-time enforcement enable generalization under limited supervision? We are actively investigating this.
> >
> > The reviewer agrees with this explanation and is happy with the clarification. I think this point could be even more strongly emphasized in the paper - in particular, in the introduction and the conclusion sections.
> >
> > >> Q3: Where is the smoothing function defined? Is it just the sum operator?
> > > Thanks for this question, since this is a key contribution of our work but, we agree its description would deserve additional space to make full justice to it. ...
> >
> > Thank you for the detailed explanation and this clarifies the question!
> >
> > >> Q4: Figure 8: Does the performance get even better when you perform more iterations/steps?
> > > Yes. By proceeding with more steps, it is in principle possible to achieve even smaller errors, as we have empirically observed. Since the computation time increases, however, it is also important to strike a good trade-off between these two desiderata.
> >
> > Thank you. I think in this case, it would actually make sense to show a Pareto front between # of iterations/computational time and the MSE such that the interested reader can take a look at the trade-off and choose the best number of iterations for their use case.

---

> > > ### Author Response · Authors · 2025-08-03
> > >
> > > Thank you for your detailed response to our rebuttal. We sincerely appreciate the feedback; this discussion has been very productive. In response to your points:
> > >
> > > > **There should also be an explanation about the meaning of the different rows in the "Structural analysis" section of Figure 4. The caption should clarify this.**
> > >
> > > Perfect; we appreciate this suggestion and indeed agree. To be specific, we will extend our description to explain that the structural analysis is derived from the external simulator and illustrates the response of the structure under progressively increasing deformation. Additionally, we will highlight the explicit connection between the stress induced by the imposed deformation illustrated in the figures and the y-axis shown in the 'Stress-strain curves' section of Figure 4.
> > >
> > >
> > > > **Can you intuitively explain here (and later in the paper) why this [the direction scaling is absorbed into the proximal update] is the case?**
> > >
> > > While the connection to standard finite differences can be explicitly drawn, our implementation is effectively a special case of the perturbed optimizer framework described by [1]. Therin, in Proposition 3.1, the gradient is implicitly scaled by $\frac{1}{\epsilon}$ as a consequence of integration of parts. Hence, this term is indeed present in the first order gradients (with a squared term $\frac{1}{\epsilon^2}$ which would appear in the second-order gradients).
> > >
> > > However, this does not appear explicitly in our updates as it is absorbed by the step size. Explicitly, the proximal update defined by the inner minimizer would become:
> > >
> > > $$
> > > z^{i+1} \leftarrow z^i - \text{lr}_\text{original} \cdot \frac{\bigl(\bar{\phi} \\, \epsilon (z^i)-\text{target}\bigr)}{\epsilon}
> > > $$
> > >
> > > This expression can be simplified by scaling the learning rate such that $\text{lr}_\text{updated} = \frac{\text{lr} \\, \text{original}}{\epsilon}$, making the expression:
> > >
> > > $$
> > > z^{i+1} \leftarrow z^i - \text{lr}_\text{updated} \cdot \\bigl(\bar\phi \\, \epsilon (z^i)-\text{target}\bigr)
> > > $$
> > >
> > > Thus, we implicitly cancel the explicit division operation by scaling the step size accordingly.
> > >
> > > Importantly, the proximal update (and more specifically the inner minimizer formula) also incorporates the distance regularization term:
> > >
> > > $$
> > > \tfrac{1}{2\lambda}\|\mathcal{D}(\mathbf{z}_t^{i}) - \mathcal{D}(\mathbf{z_t^0})\|_2^2
> > > $$
> > >
> > > As its gradient does not implicitly include the $\frac{1}{\epsilon}$ scaling, the relative weighting will shift if it is not similarly scaled, but it is handled simply by scaling $\lambda$ such that the ratio $\frac{\text{lr}_{\text{updated}}}{\lambda}$ remains consistent. As $\lambda$ is a hyperparameter that is  empirically optimized, this may not be the most essential consideration, but we will make sure to note this aspect as well.
> > >
> > > As mentioned in our previous response, we intend to add this explanation to our manuscript to fully justify our presentation.
> > >
> > >
> > > > **Limitations of training time enforcement: I think this point could be even more strongly emphasized in the paper - in particular, in the introduction and the conclusion sections.**
> > >
> > > Thank you for this suggestion! It will be done.
> > >
> > >
> > > > **It would actually make sense to show a Pareto front between # of iterations/computational time and the MSE such that the interested reader can take a look at the trade-off and choose the best number of iterations for their use case.**
> > >
> > > Thank you for this thoughtful suggestion. We agree that an illustration of the Pareto front would provide an interesting visualization of this trade-off.
> > >
> > > Indeed, the data to construct this curve is already implicitly included in Figure 4, although we acknowledge it would be valuable to include more steps beyond the four iterations. As it is our understanding that we cannot attach figures during this discussion window, we will instead include a table below which could be plotted to construct the Pareto front for the sample visualized.
> > >
> > >
> > > | ITERATION STEPS | 1     | 2    | 3    | 4    | 5    |
> > > |-----------------|-------|------|------|------|------|
> > > | MSE [MPa]       | 175.6 | 85.5 | 12.5 |  5.2 |  1.2 |
> > > | Runtime         |   30s |  60s | 120s | 150s | 180s |
> > >
> > >
> > > However, we note that this illustration will be specific to the sample shown. The convergence is largely dictated by the complexity of matching the target curve (e.g., how far out-of-distribution it falls), making it difficult to define a Pareto front that is broadly applicable to the experimental setting. With that in mind, we are open to incorporating this into Figure 4, with an explanation indicating this specific caveat.
> > >
> > >
> > > ---
> > >
> > > Thank you again for your feedback and careful attention to our work. We hope these responses have clarified the points discussed, but we are happy to engage in discussion further! Many thanks!
> > >
> > > [1] Berthet, Quentin, et al. "Learning with differentiable pertubed optimizers."

---

### Official Review · Reviewer_8ptq · 2025-07-03

**Clarity:** 3
**Significance:** 3
**Originality:** 3
**Rating:** 5
**Confidence:** 3

**Summary:**

This paper presents a method of incorporating proximal projection optimization as a guidance method into latent (stable) diffusion sampling models, such that the samples from the model obey feasibility constraints. This is an important consideration when using stable diffusion models for generating high dimensional, physically plausible samples in a variety of scientific and engineering applications, etc. The contribution is in formulation and analysis of the method, and in a thorough empirical evaluation on real high dimensional problems.

**Questions:**

Questions:
- Do all intermediate samples, $x_t$, have to obey the constraints, or just the final sample at $t=0$? Is the reason for constraining $x_t$ because it is too hard to propagate the constraints from only  $x_0$ to the latent space, or are the convergence rates worse, or is there another reason? It would seem to me that in some applications, feasibility of all $x_t$ would be a strong requirement that may be hard to satisfy.
- I’m confused by the dimensionality of $\mathbf{z}$ and the mapping $\mathbf{g}$ in Section 4.1, and 4.2. It appears that in section 4.1 we require $\mathcal{D} : \mathcal{Z} \to \mathcal{X}$ in section 4.1 for the inputs to $\mathbf{g}$, but in Section 4.2. Eqn. 6 this mapping is not used, and $\mathbf{g}$ has inputs from the latent space? Are you referring to different distances, $\mathbf{g}$ in these cases, or a different $\mathbf{z}$, and the notation has been overloaded?
- Inner and outer optimizer presentation confusing — I’m not actually 100% sure what these are referring to. Do you mean every diffusion step, $t$, the projection in (7) needs to be run, and then additionally, the equation after line 178 needs to be run for the inner Langevin sampling steps w.r.t. $i$? Or, is equation 7 implicitly being satisfied by the corrected Langevin sampling steps in the eqn after line 178? The latter seems to be implied by Algorithm 1, but it's not totally clear.
- “Choosing $\mathbf{g}$ as the indicator of a set, reproduces the familiar projection step introduced earlier” — This seems a little unclear to me, do you mean $g(y) = [[y \in C]]$? Would this then have to be a complement given the arg min?
- In Section 6.3 — do you need a projection path in, e.g. latent space like PCA? Or could you directly use the classifier and something like a Brier score as the $\mathbf{g}$? You mention this as future work in appendix A, what is stopping the implementation of it now?


Comments:
- Should there be no commas in the definition of $x_t$ before eqn. (1)?
- reminds me of “Deep declarative networks” (https://arxiv.org/abs/1909.04866) where notes in a network can be the solution to an optimization problem (implicit nodes).
- Algorithm 1: would be good to indicate $z^0_t$ comes from omitted sampling steps.

**Ethical Concerns:**

["NO or VERY MINOR ethics concerns only"]

**Final Justification:**

The authors have suitably responded to my queries, and by adding these clarifications, corrections and discussion points to the final manuscript will make for an interesting addition to NeurIPS.

**Limitations:**

Yes, the authors supply a nice comment on limitations in the Appendix.

**Quality:**

3

**Strengths And Weaknesses:**

Strengths:
- The problem is interesting, relevant an impactful — and will help allow diffusion models to generalise to tackle new, high-dimensional problems in science and engineering that have feasibility constraints.
- The contribution is clear and significant, and notable the authors supply a convergence analysis of their method, in terms of both the fidelity of the (constrained) generative distribution, and feasibility of generated samples.
- Strong empirical/experimental evaluation on interesting end relevant problems, which back up the claims of the authors.
- Clearly and well written for the most part.

Weaknesses:
- Some of the presentation of the method could be clearer to aid understanding — e.g. there is some confusing potential overloading of notation, and the algorithm could be explained more clearly. See the “questions” section for more detail on these points.

Overall I found this paper a pleasure to read and review. It is interesting, and I think has decent originality and significance. The quality and clarity is generally high, though could be better with some small additional effort on the part of the authors.

With these clarifications, I think this is a worthy submission to NeurIPS 2025.

---

> ### Author Rebuttal · Authors · 2025-07-30
>
> Thank you for reviewing our work carefully and for highlighting its strengths and novel contributions. We answer each question below. Please let us know if we can provide any further clarification.
>
>
> > **Q1:** Do all intermediate samples, $x_t$, have to obey the constraints, or just the final sample at $t = 0$?
>
> This is also the question we asked ourselves at the beginning of this investigation. In practice, only the final sample *must* satisfy the constraints, but constraining the intermediates matters.
> Indeed, already earlier work showed that enforcing post-sampling correction at $x_0$ results in substantial quality degradation because a late projection can *push away the point from the target probability density region* [1,2].
> Our Theorem 4.1 formalizes the gain: keeping $x_t$ within the constrained manifold steers the chain toward the constraint set while still converging to $p_\text{data}$ (Theorem 4.2).
>
> As you noted, early samples could be too noisy in some domains to be evaluated. We follow this general principle: apply projections as early as possible, while ensuring that the constraint is being properly evaluated. This is when the *highest sample quality is achieved* according to our observations.
>
>
> > **Q2:** I’m confused by the dimensionality of $z$ and the mapping $g$ in Section 4.1, and 4.2.
>
> Thank you for noting this! The notation was overloaded. In Equation (6), we meant a sample and penalty function operating in the same space (latent or ambient). We will replace $z_t$ with $x_t$ there and keep Equation (7) as the latent‑space generalization. This makes the ambient‑space case clear while preserving generality.
> Again, we appreciate this observation and this has been addressed on our end already.
>
>
> > **Q3:**  Do you mean every diffusion step, $t$, the projection in (7) needs to be run, and then additionally, the equation after line 178 needs to be run for the inner Langevin sampling steps w.r.t. $i$? Or, is equation 7 implicitly being satisfied by the corrected Langevin sampling steps in the eqn after line 178?
>
> At each diffusion step (the outer loop in Algorithm 1), Equation (7) is satisfied by the correction (the inner loop in Algorithm 1). The proximal mapping defined in this equation (and implemented in the inner loop) is facilitated by the "inner minimizer" -- essentially, the equation presented for the inner minimizer can be viewed as a *single unrolled step* of the iterative process used to solve the optimization defined by Equation (7). We will add verbiage to better clarify this.
>
>
> > **Q4:** “Choosing $g$ as the indicator of a set, reproduces the familiar projection step introduced earlier”
>
> To clarify, here we are referring to the case where $g$ is defined as an *indicator function*. In proximal operator, an indicator function has a very specific definition:
> $$
> g(y) =
> \begin{cases}
> 0, & \text{if } y \in C \\\\
> +\infty, & \text{if } y \notin C.
> \end{cases}
> $$
>
> Hence, the constraint violation term completely dominates the optimization whenever $y \notin C$, and the proximal map becomes:
>
> $$
> \text{prox}_{\lambda g}(x) = \underset{y \in C}{\text{argmin}} \\; \frac{1}{2 \lambda} \|y - x\|^2
> $$
>
>
>
>
> exactly the projection onto $C$. Additionally, provided this definition of an indicator function, there is no need for a complement. We hope this clarifies your question.
>
>
> > **Q5:** In Section 6.3 — do you need a projection path in, e.g. latent space like PCA?
>
> Not strictly, but it can significantly enhance the efficacy of the method. For instance, a scalar classifier output could potentially be used as a score to guide the correction, but, as we had observed in our early experiment, it can be too lossy (the information captured by the model is compressed into a single value in the final layer). We thus project the penultimate activations to PCA‑2 so the guidance uses richer features without running into high‑dimensional clustering issues. Higher‑dimensional PCA might also help, but exploring that trade‑off is left as a possible future direction.
>
>
> ---
>
> Thank you agin for your feedback! We hope these clarifications strengthen the paper and are happy to discuss any remaining concerns.
>
>
> [1] Christopher, Jacob K., Stephen Baek, and Nando Fioretto. "Constrained synthesis with projected diffusion models." Advances in Neural Information Processing Systems 37 (2024): 89307-89333.
>
> [2] Yuan, Ye, et al. "Physdiff: Physics-guided human motion diffusion model." Proceedings of the IEEE/CVF international conference on computer vision. 2023.

---

> > ### Comment · Reviewer_8ptq · 2025-08-02
> > **Thanks for the clarification**
> >
> > Thank you for the clarifying comments. I think adding some of these points to the main text or appendix will help readers better understand the method and engage with the paper. E.g. your answer to question 1 is very interesting, and clarifying your meaning of indicator, i.e. a mapping to the set $\\{0, +\infty\\}$ will aid in comprehension, etc.
> >
> > Well done!

---

> > > ### Author Response · Authors · 2025-08-02
> > >
> > > Thank you very much for the comments and support. We completely agree and will incorporate these details and the other clarifications discussed in the final version of the paper. Thank you again!

---

### Official Review · Reviewer_2mmt · 2025-07-04

**Clarity:** 3
**Significance:** 3
**Originality:** 2
**Rating:** 4
**Confidence:** 3

**Summary:**

This paper addresses a critical challenge in diffusion models, that is, their inability to strictly adhere to domain-specific constraints. Existing diffusion models explore  constrained generation where diffusion takes place in original data space, however, that does not immediately translate to latent space. The authors propose a novel approach that incorporates proximal mappings into the reverse steps of stable diffusion models, allowing the generated content to conform to complex constraints. Their method does not require retraining and demonstrates strong performance across tasks such as porous-material synthesis, meta-material inverse design, and copyright-constrained content generation.

**Questions:**

Questions Summarized in weakness

**Ethical Concerns:**

["NO or VERY MINOR ethics concerns only"]

**Final Justification:**

I am in favor of accepting the paper.

**Quality:**

3

**Strengths And Weaknesses:**

Strengths
The paper targets a practical and impactful problem. This method would improve applicability of latent diffusion models to critical domains where generation must adhere to string physical constraints.
The integration of proximal mappings with pretrained diffusion models without retraining is novel, practical an efficient, especially as it works with both convex and non-convex constraints. Authors also provie solid theoretical justification for their approach.
The approach is validated across diverse and challenging use cases, showing near-zero constraint violations and strong performance, suggesting generality and robustness

Weakness
For black-box constraints, the approach estimates subgradients via finite differences, which may be noisy or inefficient in high-dimensional settings.
However, authors compare their work with too few baselines, which limits the clarity on relative advantage.
While the method is empirically shown to work for non-convex and non-differentiable constraints, strong theoretical convergence guarantees are provided only for convex constraints.

---

> ### Author Rebuttal · Authors · 2025-07-29
>
> Thank you for your review and for highlighting the _novelty points in our work_!. We found the aspect you pointed out to be very constructive, and we address them below. Please let us know if any pending aspect remains to be clarified; we are very happy to engage further.
>
> > For black-box constraints, the approach estimates subgradients via finite differences, which may be noisy or inefficient in high-dimensional settings.
>
> The point is well taken: numerical differentiation can be costly and unstable as the problem size or simulation cost grows. Our goal, however, is **to give a general framework that can be applied to any black‑box model, even when classical differentiability fails**. Crucially, the differentiable sensitivity analysis tool provided makes it possible to insert previously incompatible simulators into the inner loop of diffusion models! We believe this is an important and exciting contribution, as illustrated by our results.
> Let us also remark that, as the reviewer may be aware, in this generative‑model context an approximate gradient is often sufficient; the stochastic nature of diffusion already tolerates noise, and even coarse gradient information still drives the sampler toward high‑quality outputs.
>
> In terms of efficiency, this can be improved on a case‑by‑case basis, e.g., batching model calls, parallel execution, GPU acceleration, but we consider these as engineering steps that are orthogonal to the main message and contribution of the paper and were not the focus of this initial study. While we have currently addressed this point in the limitation section, we will be happy to further emphasize it, should the reviewer deem it essential.
>
>
> > Authors compare their work with too few baselines.
>
> Throughout our work, we took special care in identifying what we found to be the **the strongest available state-of-the-art baselines** for each task: Christopher et al., for microstructure generation, and Bastek & Kochmann for metamaterial design. In the case of copyright-safe generation, since it has not been addressed in prior work, the best available state-of-the-art approach we found was conditional diffusion (Ho et al.). During development, we also considered other baselines (which are discussed in Appendix E), but we omitted them from the main text to keep the exposition focused on meaningful targets for improvement.
>
>
> > Strong theoretical convergence guarantees are provided only for convex constraints.
>
> This is not entirely accurate. Our convergence guarantees are provided under a $\beta$-prox-regularity assumption (in the sense of [A] Definition 13.27). This is *a relaxation of typical convexity assumptions*, which implies that for each viable point and normal direction, small perturbations still project uniquely and smoothly back to $\mathbf{C}$. Hence, the theoretical convergence guarantees extend beyond solely convex constraint sets.
>
> However, as the reviewer will surely know, guarantees for truly general non‑convex constraints remain out of reach for the optimization field at large, so proving results under prox‑regularity *should be considered a strength rather than a limitation*.
>
> ---
>
> Thank you again for your positive assessment of our work! We hope these clarifications address your concerns, and would appreciate your championing our work further. We are happy to follow-up if other questions arise. Many thanks!
>
>
> [A] Rockafellar, R. Tyrrell, and Roger JB Wets. Variational analysis. Berlin, Heidelberg: Springer Berlin Heidelberg, 1998.

---

> > ### Comment · Reviewer_2mmt · 2025-08-04
> > **Answers**
> >
> > The authors reasonably answers my questions. I maintain my score.

---

> ### Author Response · Authors · 2025-08-04
>
> Thank you! We remain available in case there may be additional points; Since the reviewers' questions have been answered, we also hope could see this reflected in the originality, quality, and novelty scores of this review. Again, thank you for your time and help.

---

### Note · Authors · 2025-08-11

We take advantage of this final note to thank the reviewer for their unanimous support for this work and summarize the (minor) adjustments we will make, based on your constructive suggestions:
1. **Methodological Clarifications:** We'll add explanation of why it is necessary to constrain intermediate samples, address overloaded notation (specifically between Equations 6 and 7 as pointed out by Reviewer 8ptq), and expand descriptions of the inner loop and outer loop based on our discussions. We'll provide definitions for the indicator function used in our proximal updates, and provide the full derivation and intuition for the smoothing function defined by $\phi_{\nu}(\mathbf{x})$. We'll also add description of why the finite-difference scaling is absorbed into the step size of the proximal update, following the description we outlined in the thread below.
2. **Presentation Improvements:** We'll adjust our figure captions (especially Figures 4 and 5) so that they are self-contained. As several reviewers missed the runtime analysis included, we'll be intentional in adding better sign-posting for this, including these in the main text if the allotted page limits allow. This will be supplemented by theoretical complexity results provided in our rebuttal. Finally, we will rename Appendix H as "Supplementary Proofs" to avoid an informal tone.
3. **Experimental Sections:** We will better highlight the additional training-time enforcement of baselines mentioned in Appendix E, by mentioning it in the main text. We will also note that the baselines selected and displayed in the main text represent the strongest methods currently known for the tasks explored. Finally, we will incorporate an expanded version of the Pareto frontier discussed for Section 6.2.
4. **Expanded Background:** We'll emphasize the significance of our proposed inference-time constraint enforcement, contrasting this to often less reliable training-time enforcement mechanisms. In particular, we will stress that inference-time enforcement enables formal guarantees at a *sample-level* and allows for generalization to constraints that are unseen at training-time.

Additionally, we will clarify several definitions (e.g., "first principles" versus "physical laws", etc.).
We appreciate the suggestions provided during this period! We believe our work can be an important step towards enabling physically-reliable generative models for real-world, scientifically-grounded applications.

Thank you!
\
-Authors

---

### Decision · Program_Chairs · 2025-09-17

**Decision:**

Accept (spotlight)

**Comment:**

This paper suggests a training-free algorithm for sampling latent diffusion models with differentiable constraints by composing a proximal projection operator to the standard diffusion sampling steps. The projection entails differentiation through the differentiable constraints and the latent decoder.

Reviewers found the problem tackled by this paper important and the solution devised is novel, practical and rather efficient. The method is both theoretically justified as well as validated in a solid experimental setup.
The criticism was focused on the relatively low number of baselines compared to, and some exposition issues that the authors have indicated that they will improve in camera ready version of the paper.